# Phage diversity mirrors bacterial strain diversity in the honey bee gut microbiota

Malick Ndiaye [1], Germán Bonilla-Rosso[1,2], Florent Mazel[1] & Philipp Engel [1] ✉

Bacteriophages (phages) play a crucial role in shaping bacterial communities. Documenting the relationship between phage and bacterial diversity in natural systems is fundamental to understand eco-evolutionary dynamics that shape community composition, such as host specificity, emergence of phage resistance and phage-driven microbial diversification. However, our current understanding of this relationship is still limited, particularly in animal-associated microbiomes. Here, we analyze paired bacterial and viral metagenomics data from the gut microbiota of 49 individual honeybees and reconstruct the phage-bacteria interaction network by leveraging CRISPR spacer matches and genome homology. The resulting interaction network displays a highly modular structure with nested phage-bacteria interactions within each module. Viral and bacterial alpha and beta diversity are correlated, particularly at the bacterial strain level and when considering the interaction network. Overall, our results suggest that the most relevant approach to study phage-bacteria diversity patterns should rely on strain-level resolution and the explicit use of the interaction network. This may explain why previous studies have obtained mixed results when testing for phage-bacteria diversity correlations. Finally, we call for further studies building up on these correlation patterns to probe the underlying mechanisms by considering both bottom-up and top-down regulatory mechanisms in microbiome assembly.

Bacteriophages (or phages) are ubiquitous and highly diverse bacterial viruses. They play crucial roles in shaping bacterial communities, both at short (ecological) and long (evolutionary) time scales[1–4]. At the evolutionary scale, phages can exert strong selective pressure on their hosts, driving bacterial evolution and diversification[5–11]. At the ecological scale, phages can modulate the biomass of bacterial populations through selective predation[12–14]. For example, it is estimated that 40% of the bacterial population is killed through phage predation every day in the ocean[15]. Moreover, since phages are dependent on their bacterial hosts for replication, theory predicts that phage diversity should be controlled by the diversity of the bacterial population, an example of bottom-up effect[16–18]. However, empirical evidence supporting this theory is mixed. While some recent studies report a strong correlation between viral and bacterial diversity[19–21], others find only a weak or no

association[22–25]. Similarly, several studies highlight discrepancies in viral and bacterial composition across sampling sites, often observing higher viral than bacterial turnover[20,21,26–29].

We hypothesize that these discrepancies originate from three limitations. First, while most studies have characterized bacteria communities at the species level (e.g., using amplicon sequencing data), we hypothesize that bacterial strain level is a more adequate resolution to study bacterial-phage dynamics. This is because of the high specificity of phages, which can cause viral communities to closely mirror the strain, but not necessarily the species-level composition of their hosts[30–32].

Second, many ecological surveys have not considered the structure of the phage-bacteria interaction networks (PBIN), which map infection relationships between phages and their bacterial hosts (i.e.,

---

[1]Department of Fundamental Microbiology, University of Lausanne, Lausanne, Switzerland. [2]Molecular Agroecology and Plant Ecology, Agroscope, Zürich, Switzerland. ✉e-mail: philipp.engel@unil.ch

which phage infects which host). Interaction networks are powerful tools for analyzing complex ecological systems and have been widely used in other domains of ecology, such as food webs, plant–pollinator systems, and microbial communities[33–35]. Analyzing the topology of these networks can reveal non-random structures such as modules, i.e., clusters of interacting species that are more connected to each other than to the rest of the network. These modules often represent groups with shared ecological or evolutionary characteristics, thereby simplifying community complexity and enabling a better understanding of local dynamics and ecological patterns[36,37]. Both theoretical and experimental studies show that PBIN are often modular, i.e., interactions occur within specific groups of phages and bacteria, with little overlap among groups[2,38–41]. Therefore, associations between phage and bacterial diversity are expected to be stronger within modules rather than between modules, but this has not been explored so far.

Third, while a few studies have compared the viral and bacterial metagenomes of environmental samples, they usually come from highly complex microbial communities where most phage-host interactions remain unidentified, or classified at broad phylogenetic levels, making it difficult to reconstruct ecosystem-wide PBINs[19,27,28,42].

In this study, we aimed to address these knowledge gaps by investigating the relationship between phage and bacterial diversity at different taxonomic levels and with explicit consideration of the PBIN structure. The western honeybee, *Apis mellifera*, represents an exciting model to fill these gaps for three main reasons. First, honeybees possess specialized gut microbial communities that are important for host health[43,44]. Second, the honeybee gut microbiota is relatively simple, as it is dominated by only eight highly prevalent bacterial genera, each comprising several species with many strains per species. Some of these strains tend to segregate into individual bees, while others seem to coexist[45,46]. Third, gut microbiota of honeybees harbor diverse phages that target all prevalent bacterial genera[30,47–50]. However, the diversity of these phages has so far been analyzed only using pooled bee gut samples, limiting our understanding of individual-level variation. To address this, the next crucial step is to investigate the associations between viral and bacterial diversity within the honeybee gut. Sequencing the entire gut of individual bees provides a comprehensive view of their microbiome, addressing the limitations of fecal sampling, which may capture only a partial snapshot of microbial diversity. This approach enables to overcome common challenges in microbiome studies, notably those encountered when studying complex mammalian gut communities.

Here, we used paired shotgun metagenomics to sequence both the viral and bacterial fractions of the gut microbiota of 49 individual honeybees of *A. mellifera*. We then inferred an ecosystem-wide interaction network linking the core bacterial members of the honeybee gut microbiota to their viruses. Our analysis revealed that the honeybee phage-bacteria interaction network (PBIN) is highly modular with a nested host range within modules. Viral and bacterial diversity within modules significantly correlated both within and across samples. Crucially, these correlations were the strongest at the bacterial strain-level composition within individual bees. These findings underscore the importance of strain-level interactions in shaping microbial community dynamics.

## Results

### Paired shotgun metagenomics recovers a high quantity of viral and bacterial genomes from the gut of individual honeybees

We sampled a total of 49 adult worker bees from two hives of *A. mellifera* at the University of Lausanne (Switzerland) (Supplementary Data 1). The hindgut of each honeybee was individually homogenized, and virus-like particles (VLPs) were separated from bacterial and host cells through a series of centrifugations and filtrations. The recovered DNA was subjected to shotgun metagenomic sequencing, yielding an average of 69.40 (±16.88) and 9.22 (±3.34) million reads per sample for the bacterial and viral fractions, respectively (Supplementary Data 2 and Supplementary Fig. 1). Read mapping corroborated the enrichment of bacterial and honeybee reads in the bacterial fraction, while the viral fraction was dominated by sequences matching viral genomes obtained from the honeybee gut in previous studies Fig. 1A, B, see "Methods").

From the bacterial metagenomes, we reconstructed a total of 478 bacterial metagenome-assembled genomes (bMAGs), including 330 high-quality (completeness > 90%, contamination < 5%) and 148 medium-quality genomes (completeness ≥ 75%, contamination < 10%). This dataset was further supplemented with 220 reference bacterial genomes isolated from various bee species (Data S3 and S9). These bMAGs and isolate genomes were clustered at 95% average nucleotide identity (ANI) into species-level bacterial Operational Taxonomic Units (bOTUs). To determine the absence/presence of these bOTU across our samples, we mapped the bacterial metagenomic reads against a database containing one representative genome per bOTU. On average, 80.8% (±8.9 SD) of the reads (after filtering out honeybee reads) mapped against the bacterial database, suggesting that the assembled MAGs captured most of the diversity present in the bacterial fraction. We detected 53 bOTU across the 49 individual honeybees. Of these, 36 belonged to the most prevalent genera of the honeybee gut microbiota (*Gilliamella*, *Lactobacillus*, *Bombilactobacillus*, *Bifidobacterium*, *Snodgrassella*, *Commensalibacter*, *Frischella*, and *Bartonella*) and contained 85% of the medium- to high-quality bMAGs (Supplementary Data 3 and Supplementary Fig. 2). Of these, 23 were composed of both bMAGs and isolate genomes, eight were composed of only bMAGs, and five were composed of only isolate genomes (Fig. 1C and Supplementary Data 3). Therefore, our dataset captured the majority of diversity observed in the isolate genomes, while also identifying the presence of previously undescribed species (Supplementary Data 3).

To assess viral diversity, we first predicted viral contigs from both the bacterial and viral metagenomes using a combination of different tools (see "Methods"). This resulted in the detection of 10,021 viral metagenome-assembled genomes (vMAGs) (Supplementary Data 4). The bacterial and viral fractions contained a similar number of vMAGs classified as temperate phages, with 1685 and 1682 vMAGs, respectively (Supplementary Data 4). In contrast, the viral fraction harbored more vMAGs classified as virulent phages (4103) compared to the bacterial fraction (2551; Supplementary Data 4). Notably, the viral metagenomes were significantly enriched in VLPs, while the bacterial metagenomes were enriched in prophages (Fig. 1C, D). These observations corroborate the successful enrichment of viral sequences in the viral fraction, specifically capturing VLPs rather than viral sequences integrated into bacterial host genomes.

We clustered the vMAGs at 95% ANI and 85% alignment fraction[51] ("Methods") and obtained 1'069 viral OTUs (vOTUs) having at least one vMAG of medium-quality or higher according to CheckV (Supplementary Fig. 3). As for the bacterial fraction, to determine the absence/presence of these vOTU across our samples, we mapped the viral metagenomic reads against a database containing one representative genome per vOTU (see "Methods"). This resulted on average in 87.7% (±8.4 SD) mapped reads per sample with 937 vOTU detected across the 49 individual bee samples, showing that we captured the vast majority of the diversity present in the viral metagenomes.

In summary, these results show that it is possible to comprehensively assess the diversity of virus-like particles and gut bacteria from individual bee guts using shotgun metagenomics.

### Phage-bacteria interactions are conserved, highly modular and nested within modules

To establish a phage-bacteria interaction network (PBIN) for the bee gut microbiota, we inferred which phages interact with which bacteria using CRISPR spacer matches and genome homology (Supplementary

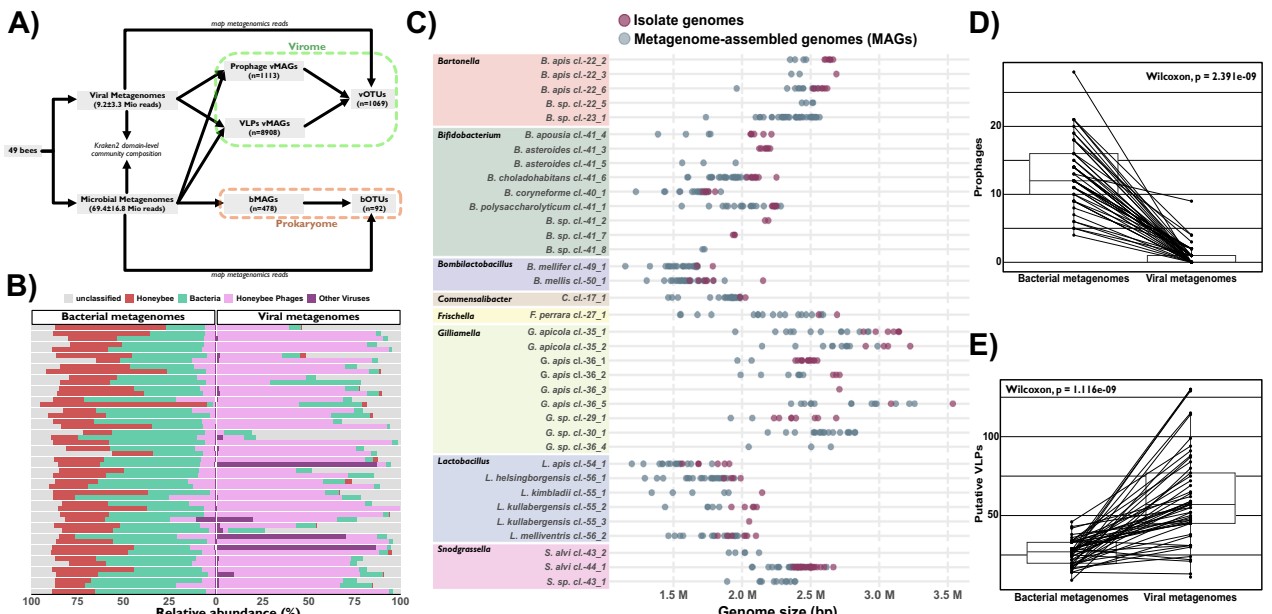

**Fig. 1 | Reconstruction of bacterial and viral genomes using a paired metagenomics approach on 49 individual honeybee gut microbiomes.** Source data are provided as a Source Data file. **A** Overview of the bioinformatics pipeline used to analyze the paired metagenomes. **B** The percentage of reads mapped to the genomes of organisms expected in the bacterial (left) and viral (right) metagenomes using Kraken2. **C** Genome sizes of the recovered bMAGs and isolate genomes of different bacterial species belonging to the core genera of the honeybee gut microbiota (rows). **D**, **E** Boxplots comparing the number of prophages (**D**) and putative VLPs (**E**) recovered in the viral and bacterial metagenomes ($n = 49$). Boxplots indicate the median (centre line), the 25th and 75th percentiles (bounds of the box), and whiskers extending to $1.5 \times$ the interquartile range. All individual data points, including values outside the whiskers, are shown. Two-sided paired Wilcoxon signed-rank test $p$-values are reported on the plot.

Data 5; see "Methods"). Through these two approaches, we linked 75.2% of the vMAGs to at least one bacterial genome Fig. 2B, Supplementary Data 6 and Supplementary Fig. 4D), thereby assigning hosts to most identified vMAGs. Among the 937 medium- to high-quality vOTU detected in the 49 individual bees, 616 (66%) were predicted to infect core honeybee bacteria. An additional 97 vOTUs (10%) were assigned to non-core bacterial hosts, while 224 (24%) could not be assigned a host based on our database. vOTUs lacking host predictions or associated with non-core bacteria were excluded from downstream analyses to focus on phage–host interactions within the core gut community. Notably, *Snodgrassella*, *Frischella*, nor *Bombilactobacillus* do not harbor CRISPR systems in their genomes. As a result, interactions involving these three genera were inferred solely through genome homology with prophages integrated in the bacterial genomes, which might have limited power to detect all interactions. Furthermore, to ensure that CRISPR arrays were not mistakenly assigned to the wrong bMAG, which would result in wrong bacteria-phage matches, we compared the number of spacers recovered in the bMAGs with the ones recovered from isolate genomes. Genomes from different genera had a significantly different number of spacers (Kruskal-Wallis, $p < 0.001$; Supplementary Fig. 4B). However, no significant differences were found between bMAGs and isolate genomes of the same genus (Supplementary Fig. 4C), suggesting accurate CRISPR array binning and hence phage-bacteria matching.

Based on the CRISPR spacers and genome homology matches between bacterial genomes (bMAGS and isolates) and vOTUs, we then established the PBIN (Fig. 2B). Using a modularity optimization algorithm, we found that the PBIN was highly modular ($Q = 0.71$, where Q ranges from 0 to 1, with higher values indicating greater modularity) and identified nine interaction modules (IMs; see "Methods").

All IMs corresponded to monophyletic groups of bacteria, mostly coinciding with the core genera of the bee gut microbiota (Fig. 2A), suggesting that the modularity in the honeybee PBIN is explained by

bacterial phylogenetic relationships. Only 23 vOTUs exhibited genome homology with bacteria from more than one genus (Supplementary Data 6). Of these, 16 vOTUs were associated with *Frischella* and *Gilliamella*, consistent with their clustering within the same IM (Fig. 2B). The remaining 7 vOTUs showed genome homology with *Frischella*, *Gilliamella*, and *Snodgrassella*, which is biologically plausible given the close phylogenetic relationship among these genera (Fig. 2A). Overall, this finding suggests that phages with broad cross-genus host ranges are relatively rare in the honeybee gut virome. A similar observation was made for the phages. vOTUs from the same IM clustered together based on the fraction of shared proteins as inferred by vConTACT v2.0 (Fig. 2C and Supplementary Fig. 11; see "Methods").

Previous studies have shown that bacteria-phage interaction networks typically exhibit a nested structure, where some phages interact with a broad range of bacteria, while others are more specific to only a subset of the bacteria infected by the generalist phages. To test for nestedness within the IMs of the bee gut microbiota, we compared the observed nestedness metric value for each IM (NODF[52]) with the nestedness metric values of a null-model matrix (i.e., randomized matrix, see Methods), revealing that seven (*Gilliamella_Frischella_1*, *Snodgrassella_3*, *Bartonella_5*, *Bartonella_6*, *Bifidobacterium_7*, *Lactobacillus_8*, and *Bombilactobacillus_9*) of the nine IMs were significantly nested (Fig. 2D). The *Snodgrassella_3* and *Commensalibacter_4* IMs had the fewest phage–bacteria pairs, which may explain why they did not exhibit statistically significant nestedness. To ensure our results were not artefacts of the bMAGs, we also reconstructed the PBIN using only genomes from isolate bacteria, which resulted in the same modular-nested structure (Supplementary Fig. 5).

Collectively, these findings highlight that the interaction network between phages and bacteria in the bee gut is modular and nested within modules and that the modularity is driven by congruent genetic units at the level of both the bacterial hosts and their infecting phages.

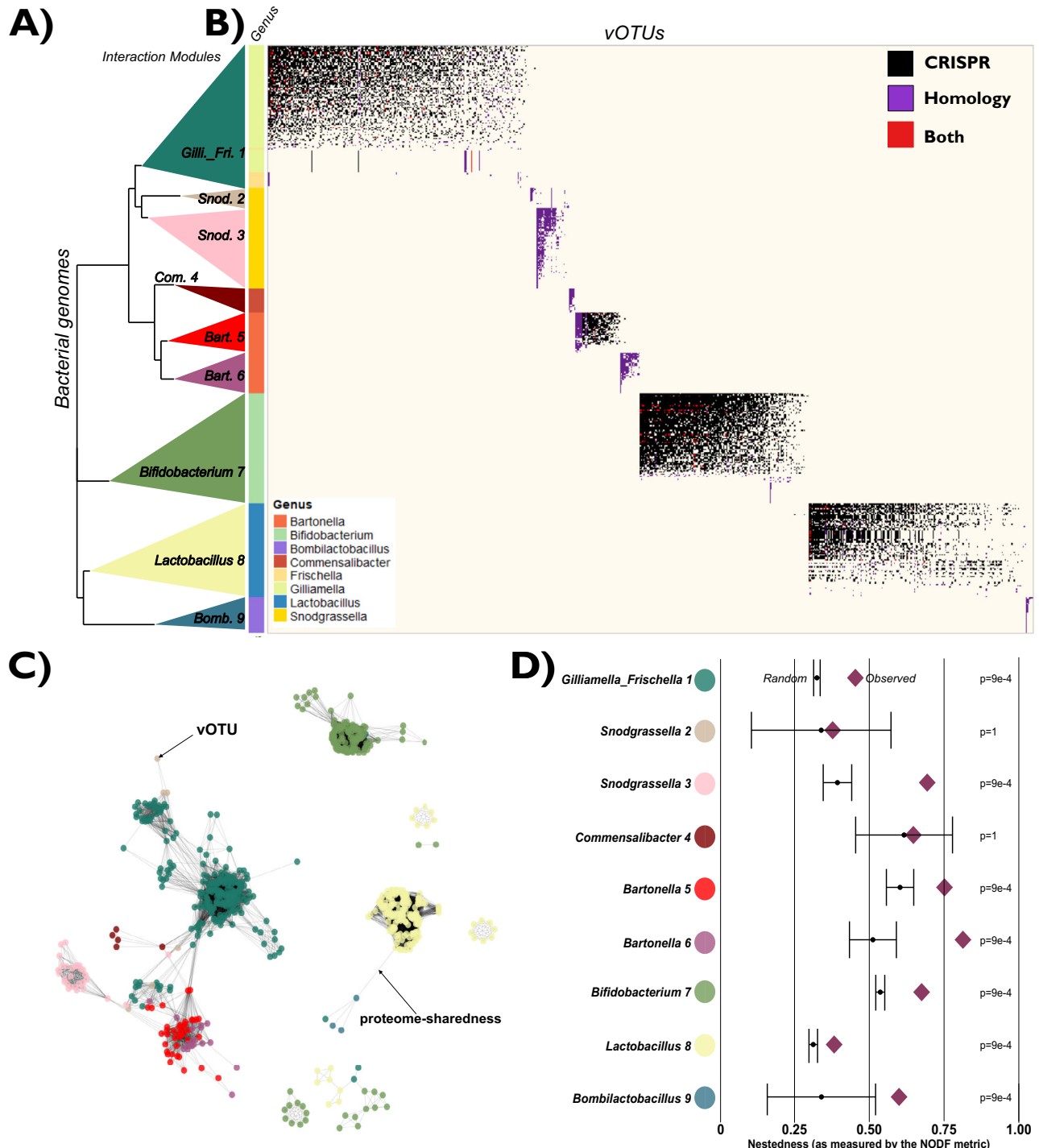

**Fig. 2 | Modularity in the phage-bacteria interaction network (PBIN) reflects genetically congruent units.** Source data are provided as a Source Data file. **A** Bacterial cladogram based on 101 concatenated orthogroups, with interaction modules collapsed into colored triangles. **B** Matrix showing predicted phage-bacteria interactions (bMAGs and isolates vs. vOTUs), ordered to maximize nestedness and by interaction module (IM). Dot colors indicate prediction methods. Bacterial genera are shown by colored bars beside the rows. **C** Proteome-sharedness network among vOTUs, with nodes representing vOTUs, edges indicating shared proteins, and colors corresponding to IMs. Only vOTUs assigned to an IM are shown. **D** nestedness metric based on overlap and decreasing fill (NODF) for each IM. A NODF value of 1 indicates maximum nestedness, while a value 0 indicates minimum nestedness. Diamonds represent observed values, while black dots with error bars represent mean NODF and 95% confidence intervals estimated from 1000 null models. Statistical significance was assessed using two-sided comparisons of observed values against the null distribution (Bonferroni correction for multiple comparisons).

## Viral composition mirrors bacterial strain-level composition across individual bees

If the identified phage-bacteria interaction modules are ecologically relevant, we expect to observe strong correlations between the bacterial and viral community structures within individual bees.

First, we noted that total DNA yield in the viral fraction correlated with the total number of 16S rRNA gene copies detected in the bacterial fraction using qPCR (Pearson's $R = 0.47$, $p < 0.001$; Supplementary Fig. 6A), suggesting that viral and bacterial biomass are linked across bees.

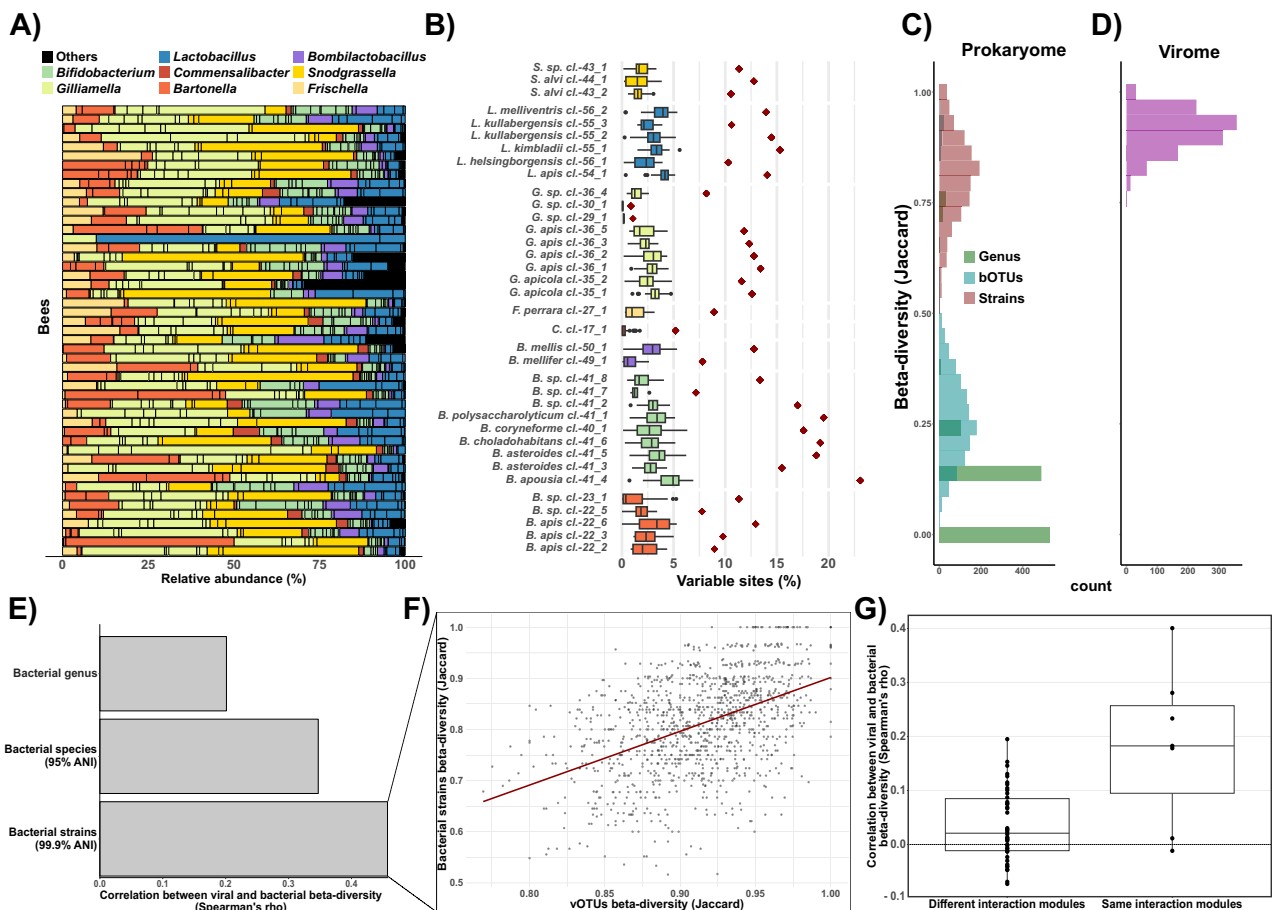

**Fig. 3 | Virome composition is correlated with strain-level composition of bacterial hosts.** Source data are provided as a Source Data file. **A** Bacterial community composition of the gut microbiota of 49 bees, with bars colored by genus. Vertical black lines separate bOTUs. **B** Percentage of variable sites (*x*-axis) detected across the genome of each bOTU (*y*-axis). Boxplots show variation across individual bees, while diamonds represent totals across all 49 bees. Boxplots show the median (centre line), the 25th and 75th percentiles (bounds of the box), the whiskers extend to 1.5 × interquartile range, and points beyond are plotted as outliers. **C, D** Jaccard dissimilarity distributions for bacterial communities (**C**) at genus, species, and strain (99.9% popANI) level, as well as for (**D**) viral communities (vOTU level).

**E** Coefficients of the correlation (Spearman's rho) between viral Jaccard compositional dissimilarity and bacterial Jaccard compositional dissimilarity at genus, species, and strain level. **F** Correlation between viral and strain-level bacterial beta-diversity. **G** Boxplot of Spearman's rho coefficients comparing the correlation between viral and strain-level bacterial Jaccard compositional dissimilarity, when bacteria and phages belong to the same (*n* = 7) or different (*n* = 42) interaction modules. Boxplots indicate the median (centre line), the 25th and 75th percentiles (bounds of the box), and whiskers extending to 1.5 × the interquartile range. All individual data points, including values outside the whiskers, are shown.

Next, we documented the composition of the viral and bacterial communities across bees (beta-diversity). We found that the bacterial communities were quite similar across individual bees, both at the genus and the species (i.e., bOTU) level, resulting in low beta-diversity values Fig. 3A, C; mean genus-level Jaccard distance = 0.11 ± 0.15; mean bOTU-level Jaccard distance = 0.29 ± 0.16). In contrast, at the strain level we found high variability: the fraction of single nucleotide variants (SNVs) per bOTU ranged from 0–7%, while the total fraction of SNVs across the 49 bees was considerably higher, ranging from 1–23% (Fig. 3B). Correspondingly, beta-diversity values based on the presence/absence of genomes with ≥ 99.9% ANI across individual bees were much higher (mean strain-level Jaccard distance = 0.81 ± 0.09) than for the genus or species level (Fig. 3C; "Methods"). This indicates that individual bees harbor only a subset of the strains found across bees.

Phages of the major IMs were found in almost all bee samples, notably phages of the IMs *Bifidobacterium_7*, *Lactobacillus_8,* and *Gilliamella_Frischella_1* were consistently present (Supplementary Fig. 6B). In contrast, at the vOTU-level the viral communities of individual bees were highly variable (mean vOTU-level Jaccard distance = 0.91 ± 0.04; Fig. 3D), mirroring the variability found at the bacterial strain level. Within a given IM, the host range of vOTUs was positively

correlated with their prevalence across bees (Supplementary Fig. 6C). This pattern suggests that nestedness within IM could reflect broader distribution patterns, where generalist phages (with more hosts) are more widespread across bees, while specialists tend to have a more limited and nested distribution.

To test if the viral and bacterial community compositions are linked, we used Mantel tests using Jaccard distances. Dissimilarities in composition between bacteria and phages in individual bees showed a significant correlation, with the highest correlation observed at the strain level of bacterial composition, the lowest at the genus level, and an intermediate correlation at the species level (Fig. 3E, F, Supplementary Fig. 7A and Supplementary Data 10). Critically, we found that these correlations hold when comparing viruses and phages belonging to the same IM, but not different IM (Fig. 3G, Supplementary Fig. 7B and Supplementary Data 7).

Collectively, these results suggest that the bee virome is highly variable among individual bees, with this variation best explained by strain level differences in bacterial composition. Moreover, our results show that the similarity between bacterial strain composition and viral composition is driven by the modular-nested structure of the PBIN, suggesting that they present ecologically relevant units.

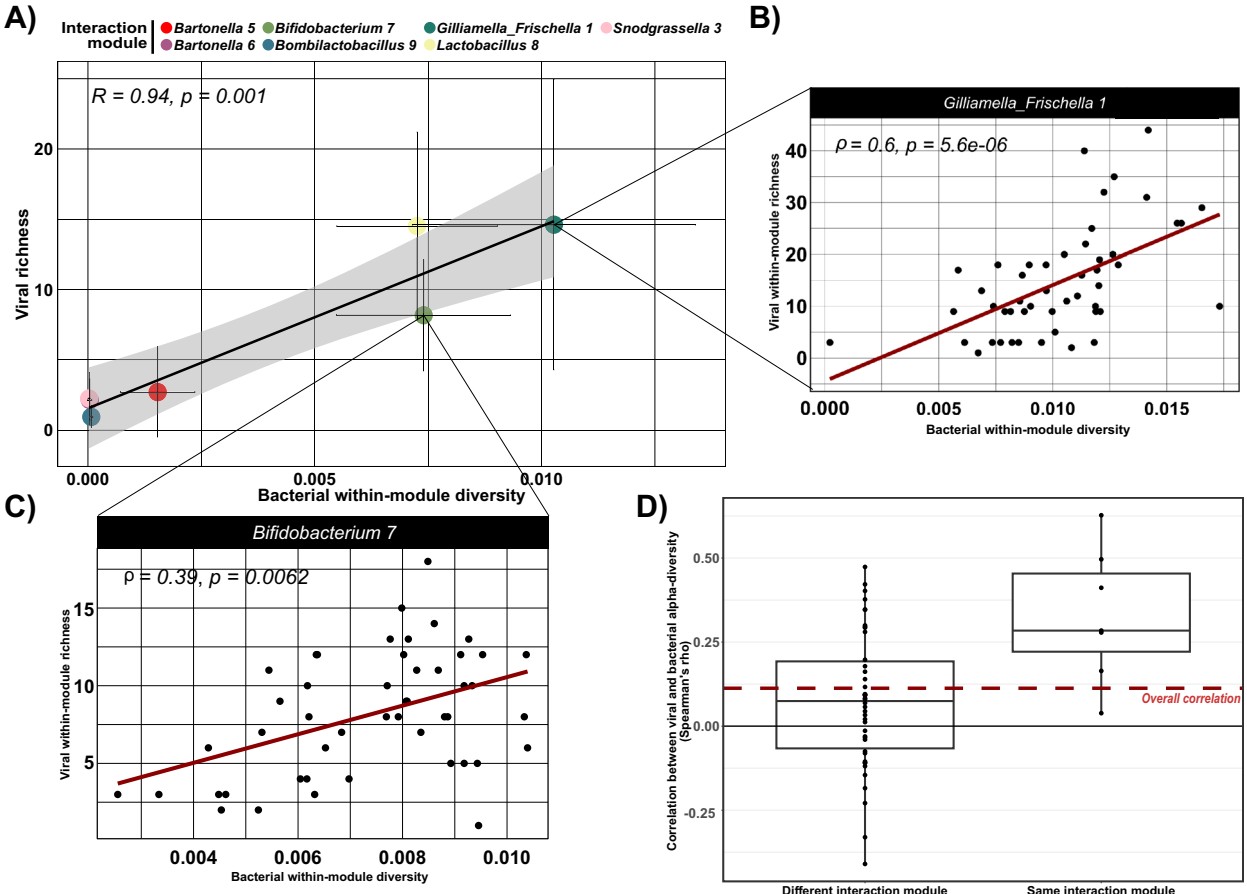

**Fig. 4 | Viral and bacterial alpha diversity correlates at different scales.** Source data are provided as a Source Data file. **A** Correlation between bacterial within-module diversity (phylogenetic diversity × average nucleotide diversity) and within-module viral richness (vOTUs counts). The color-coded dots represent interaction modules (IMs) where both bacteria and phages were identified in a minimum of 15 bees. Dots represent mean values across bees, while standard deviation is represented by the error bars. Pearson's correlation coefficient (R) and associated two-sided correlation test p-value are displayed in the plot. **B, C** Correlation between bacterial within module diversity (phylogenetic diversity

× average nucleotide diversity) and viral richness for (**B**) *Gilliamella_Frischella 1* and (**C**) *Bifidobacterium 7* IMs. Spearman's coefficient (rho) and associated two-sided correlation test p-value are displayed in the plot. **D** Boxplot of Spearman's rho coefficients comparing bacterial average nucleotide diversity and viral richness correlations when bacteria and phages belong to the same ($n = 7$) or different ($n = 42$) interaction modules. The dashed line represents the rho when ignoring the PBIN structure. Boxplots indicate the median (centre line), the 25th and 75th percentiles (bounds of the box), and whiskers extending to 1.5 × the interquartile range. All individual data points, including values outside the whiskers, are shown.

## Diversity begets diversity in phage-host interactions

To test if the PBIN is also key to detect correlations between alpha-diversity (e.g., richness) of phages and bacteria, we carried out correlation analysis both between and within modules. For the between-module analysis, we developed a metric for each interaction module (IM) that integrates bacterial species diversity, measured as phylogenetic diversity (PD), with average nucleotide diversity at the strain level. We refer to this combined measure as bacterial within-module diversity (see "Methods"). This metric reflects the idea that overall strain diversity is shaped both by the number and phylogenetic divergence of species (captured by PD) and by the genetic variability among strains within each species. By multiplying PD by the average nucleotide diversity of the species in an interaction module (IM), we obtain a composite measure that approximates the total bacterial genetic diversity within that IM in a sample. Viral within-module diversity was expressed in terms of the number of vOTUs per IM per bee (richness). We found a strong correlation between the two measures (Pearson's $R = 0.94$, p-value = 0.001; Fig. 4A), i.e., on average, IMs that held more bacterial diversity per bee also encompassed a larger diversity of phages per bee. This correlation could not be attributed to differences in sequencing depth among the samples, as we detected no significant increase in the number of bOTUs and vOTUs in function of sequencing depth (Supplementary Fig. 8A, B).

For the within-module analysis, we observed that bees with higher bacterial within-module diversity for a given IM also harbored a greater number of predating phages. This correlation was statistically significant in 4 (*Gilliamella_Frischella_1*, *Bartonella_5*, *Bifidobacterium_7*, and *Lactobacillus_8*) out of 7 IMs (Fig. 4B, C and Supplementary Fig. 9) where both phages and bacteria were detected in at least 15 samples (*Gilliamella_Frischella_1*, *Snodgrassella_3*, *Bartonella_5*, *Bartonella_6*, *Bifidobacterium_7*, and *Lactobacillus_8*, *Bombilactobacillus_9*).

To determine the bacterial genetic resolution at which phage-bacteria diversity correlations are best detected, we performed multiple linear regressions at different bacterial genetic resolutions. We found that average nucleotide diversity consistently exhibited a significant positive slope, whereas the slope for species diversity was not significant (Table 1). This was robust to variation in sequencing depth (Supplementary Fig. 10). These findings suggest that strain-level diversity is a stronger predictor of viral richness across bees than species diversity.

Finally, the relationship between average nucleotide diversity and viral richness was best explained by the structure of the PBIN. Correlations between average nucleotide diversity and viral richness were significantly stronger when phages and bacteria came from the same IM than when considering those from different IMs or when considering the communities of individual bees as a whole (Fig. 4D and

**Table 1 | Correlation between bacterial species diversity or average nucleotide diversity and viral richness**

| IM | Species diversity (PD) | | | | Strain level diversity (Average Nucleotide Diversity) | | | |
|---|---|---|---|---|---|---|---|---|
| | Slope | SE | z | P | Slope | SE | z | P |
| *Gilliamella_Frischella_1* | 2.4271 | 13.5275 | 0.1794 | 0.8576 | 2124.753 | 892.6581 | 2.3803 | 0.0173 |
| *Bartonella_5* | 18.9329 | 12.6802 | 1.4931 | 0.13541 | 203.7033 | 82.1222 | 2.4805 | 0.01312 |
| *Bifidobacterium_7* | 14.8004 | 18.4318 | 0.803 | 0.42198 | 482.9583 | 191.702 | 2.5193 | 0.01176 |
| *Lactobacillus_8* | 18.686 | 13.841 | 1.35 | 0.177 | 919.3219 | 221.5126 | 4.1502 | 3.3e-05 |

Results of the linear model show a statistically significant positive effect of average nucleotide diversity, but not species diversity, on viral richness (vOTUs count). Each row reports the effect size of the two-diversity metrics on viral richness (slope), standard error (SE), as well as the Wald z-statistic for its effect size and the corresponding two-sided p-value.

Supplementary Data 8). These results highlight the importance of considering the structure of the PBIN to uncover patterns of ecological diversity.

## Discussion

Here, we hypothesized that correlations between phage and bacterial composition and diversity will be revealed when explicitly considering the PBIN structure along with the use of adequate bacterial genetic resolution. Using paired viral and bacterial shotgun metagenomics of 49 gut samples from individual honeybees, we confirm these predictions and highlight the importance of host range and strain level interactions in shaping microbial community structure. These findings provide insights into the dynamics governing bacterial and viral diversity in animal-associated microbial ecosystems.

To identify bacterial hosts for most recovered vMAGs and build the PBIN, our study leveraged the historical genomic imprints of past infections using CRISPR spacer-to-protospacer matches and genome homology. The PBIN modularity was strongly explained by the genetic makeup of bacteria and phages. In other terms, genetically similar phages tended to interact with several genetically similar host species, most often from the same bacterial genus. Indeed, the *Commensalibacter_4*, *Bifidobacterium_7*, *Lactobacillus_8*, and *Bombilactobacillus_9* IMs encompassed all bacterial species from their respective genera. These findings are supported by studies where phages of *Staphylococcus* and *Agrobacterium* have been shown to possess broad and modular host ranges, which are best explained by the phylogenetic relationships among the bacteria[53,54]. Fundamental differences in membrane receptors, transcription–translation machinery, and general metabolism between bacteria are likely explanations for why these interaction modules are phylogenetically restricted[2]. However, genus boundaries did not always define distinct interaction modules (IMs) in our study: bacteria from two related genera, *Frischella* and *Gilliamella*, belonged to the same IM, while those of the genera *Bartonella* and *Snodgrassella* were associated with two distinct IMs (Fig. 2). Which factors determine such differences in host range remains to be understood. However, it is notable that *Frischella* and *Gilliamella* are metabolically similar and colonize the same niche in the bee gut[44,55,56], which may explain their shared phage interactions. Conversely, the genera *Bartonella* and *Snodgrassella* were each divided into two distinct IMs exhibiting high phylogenetic divergence (Fig. 2A), suggesting that substantial genetic differences within a genus can lead to the formation of separate IMs. Future studies on the strains associated with these IMs will be valuable for elucidating the eco-evolutionary factors that drive IM affiliation, such as differences in spatial localization within the honeybee gut or variations in membrane receptors.

Within modules, phage-bacteria interactions consistently exhibited a nested structure in the bee gut microbiota. This pattern appears to be a conserved feature of PBINs across ecosystems. It has been observed in experiments with both *E. coli* and *Vibrio*, along with their corresponding phages, as well as in an ecological survey of *Sulfolobus islandicus* and *Pseudomona aeruginosa* and their phages[38,39,41,57,58].

Nestedness can arise because of the coevolution between bacteria and their predatory phages[57,59] and it has been linked to the maintenance of diversity in environments with multiple bacterial strains and phages coexisting[60]. Moreover, we found a positive correlation between phage host range and prevalence across bees, with generalist phages (infecting more bacterial strains) being more prevalent than specialist phages. This aligns with observations from ecological networks in other systems[61–63]. Whether the observed nestedness is primarily driven by neutral ecological patterns, by evolutionary coevolutionary dynamics, or by an interplay of both remains an open question. Given the experimental tractability of the bee system, future longitudinal and manipulative studies could help to start disentangling these contributions.

Intuitively, the high specificity of phage-bacteria interactions should result in strong correlations between viral and bacterial community compositions (as reflected by beta-diversity) as well as between viral and bacterial alpha-diversity[17,18]. However, empirical evidence of these patterns is so far mixed. Viral beta-diversity often exceeds bacterial beta-diversity across ecosystems, and correlations between the two vary in strength and significance[20,21,23,26–29,64]. Likewise, some of the same studies report correlations between viral and bacterial alpha-diversity at the ecosystem level[19–21], while others find weak or no significant relationship[22–25].

We observed high viral beta-diversity among individual bee guts, in striking contrast to the well-documented relative stability and conservation of the honeybee bacterial community at the genus and species level[43,44,46]. Conversely, overall levels of bacterial strain and viral beta-diversity were similar, and beta-diversity values strongly correlated across pairs of samples at the strain level. Several experimental studies, including one from the bee gut, have shown that phages can infect several bacterial species, but that they often target only a subset of strains within a species[30,39,65]. We propose that the stronger strain-level correlation reflects how viral populations interact with specific sets of strains across species, rather than with distinct species. In turn, this shows how ecological patterns can be used to infer the relevant taxonomic resolution at which phage-bacteria interactions occur in natural communities.

Our results also show that interaction modules with more genetically diverse strains support a more diverse viral community. Similarly, variation in viral diversity across bees was strongly correlated with variation in strain-level diversity of the bacteria. Notably, this alpha-diversity correlation weakened when the PBIN structure was ignored, underscoring the importance of host range in shaping phage-bacteria dynamics.

Both beta- and alpha-diversity correlations tended to be weaker for IMs composed of bacterial species lacking CRISPR-Cas systems (*Snodgrassella_2*, *Snodgrassella_3*, *Bombilactobacillus_9*, and *Bartonella_6*; Supplementary Fig. 7b and Supplementary Fig. 9). This pattern may reflect an inability to capture the full diversity of their associated phages, as we were unable to use CRISPR spacer matches to link phages to their bacterial hosts.

Based on these findings, we suggest that PBIN reconstruction and integration in the diversity analysis, along with measuring diversity at the strain-level, will be key for future studies to unravel universal diversity patterns in phage-bacteria interactions across the ecosystem. While not explicitly tested for, our findings also suggest that the type of phages recovered from bacterial metagenomes and VLP metagenomes are largely different, calling for more studies applying paired bacterial and VLP metagenomic approaches. Moreover, our study was limited to two hives from the same apiary in Lausanne (Switzerland). Future research should expand sampling to more hives across diverse geographic locations, both to validate our findings and to explore additional factors that may influence the topology of the PBIN and the diversity correlations, such as spatial proximity between bacterial strains and phages.

Documenting correlations in diversity between phage and bacteria across systems is the first step to establish broad and consistent ecological patterns in nature. In turn, these patterns provide the foundation for investigating the eco-evolutionary dynamics of phage-host interactions and their impact on ecosystems and host health. However, correlations can emerge from multiple mechanisms, and the next step will be to decipher whether the reported correlations emerge because of the bottom-up effect (bacteria diversity driving phage diversity), the top-down effect (phage diversity driving bacteria diversity), or a mix of both.

If bacterial diversity drives phage diversity, it remains to be determined what influences different strains to co-exist or segregate into individual hosts. In the case of the bee gut microbiota, differences in strain-level diversity may be due to variation in diet and hence the physicochemical gut environment, e.g., between forager and nurse bees[66]. Another important process may be priority effects, where the order and timing of bacterial colonization in newly emerged bees, rather than niche or fitness differences - create individualized strain profiles[46,67]. Alternatively, phages may exert top-down control by preventing competitive exclusion among bacteria through mechanisms like kill-the-winner, which balance competition and promote coexistence of several strains in the same gut[18,60,68]. Similarly, phages have been shown to have a strong impact on the assembly of microbial communities on particulate organic matter in the sea[69]. Thus, phage predation in the early stages of bee gut colonization may significantly influence which strains establish within a given bee. Top-down and bottom-up mechanisms are not mutually exclusive and may jointly maintain diversity within microbial communities. However, there is a notable lack of studies that experimentally manipulate both top-down (i.e., phage predation) and bottom-up (e.g., nutrient availability) to quantify their individual and combined effects on community diversity[16,70,71]. Understanding the relative contribution of these two effects is essential to critically evaluate the role of phages in shaping host-associated microbiomes and our study highlights possible ecological drivers (strain-specificity and structure of interactions) that should be tested in future experiments. We suggest that this experimental work will be essential to disentangle who drives whom. Due to its experimental tractability, the bee microbiota offers not only a suitable model to test these alternative scenarios, but also to assess how phages impact the bacterial community composition in the gut of these important pollinator species and hence provide insights into bee health.

## Methods

### Sampling & DNA extraction
We collected 49 adult female worker bees of *A. mellifera* from 2 hives in Lausanne (Switzerland) in spring 2021 (Supplementary Data 1). The bees were anesthetized using $CO_2$ and put on ice. Then, they were individually dissected to extract the entire hindgut (pylorus, ileum, rectum). Each hindgut was placed in a tube containing beads (0.75–1 mm glass beads; Carl Roth) and sterile SM buffer (200 mM

NaCl, 10 mM MgSO4, 50 mM Tris-HCl, pH 7.5, 0.01% gelatin). Samples were homogenized at 6 m/s for 40 sec in a Fast-Prep24 5 G homogenizer (MP Biomedicals). Following centrifugation (17'000 × *g* for 3 min at 4 °C), the pellet was stored at − 20 °C for later DNA extraction of the bacterial fraction. The supernatant was recovered and sequentially filtered through a 0.45 µm cellulose acetate filter and a 0.22 µm centrifuge tube filter (Corning Costar Spin-x). Then, the filtrate was treated with DNase I and RNase A (Sigma-Aldrich) for 1 h30 min at 37 °C to degrade free nucleic acids not protected by capsids. The nucleases were inhibited using a lysis buffer (EDTA 0.5 mM, Tris 1 M and ddH2O) and incubation at 65 °C for 10 min. Next, VLPs were lysed by one incubation with 75 µL SDS 10% and 12.5 µL proteinase K (20 mg/mL).

The pellets from the bacterial fraction were thawed at room temperature. Then, bacterial cells were resuspended in 1X PBS, 2X CTAB, 2 µL ß-mercaptoethanol, and 20 µL proteinase K (20 mg/mL). Next, 0.1 mm Zirkonia/Silica beads were added to the solution and cells were lysed at 6 m/s for 40 sec in a Fast-Prep24 5 G homogenizer (MP Biomedicals) followed by incubation at 56 °C for 1 h.

Both DNA from the bacterial and phage fractions was extracted with a phenol/chloroform/isoamyl alcohol protocol (Sigma-Aldrich), followed by precipitation with 70% ETOH and linear polyacrylamide (Sigma-Aldrich) overnight, as well as washings with EtOH. The DNA was finally eluted in nuclease-free water and submitted to a further clean-up using DNA-specific magnetic beads (clean NGS).

The total 16S rRNA gene copy number for the bacterial fraction was assessed using qPCR with a set of universal 16 primers (F: AGGATTAGATACCCTGGTAGTCC; R: YCGTACTCCCCAGGCGG) generated in Kešnerová et al., 2017.

### Shotgun metagenomics sequencing
Libraries were prepared with the Illumina Nextera Flex library kit (Illumina) with unique dual indices (UDI) and sequenced on an Illumina NovaSeq 6000 instrument (PE150) at the Genomic Technologies Facility (GTF) of the University of Lausanne. The quality of the reads was assessed with FastQC (v0.11.4, Babraham Institute) and subsequently trimmed and filtered to remove low quality sequences and short reads using the tool Trimmomatic v0.35[72] with settings PE ILLUMINACLIP:NexteraPE-PE.fa:2:30:10 LEADING:28 TRAILING:28 MINLEN:40. Following low-quality reads removal, reads mapping to the *A. mellifera* and the human genome were removed using bbsplit v38.18[73].

### Bacterial and viral fraction enrichment validation
To initially assess the enrichment of viral sequences in viral versus bacterial metagenomes, raw shotgun reads from both fractions were mapped against a custom Kraken2 database. This database included bacterial and viral RefSeq genomes libraries, phage genomes from Bonilla-Rosso et al. (2020), and the *Apis mellifera* Amel_HAv3.1 reference genome[74,75]. The relative abundances of these genomes were determined based on the proportion of reads mapped to each group.

### Bacterial Isolate Genomes Recovery, Sequencing and Assembly
A total of 220 bacterial isolate genomes were used in this study. Of these, 211 genomes were retrieved from an in-house database consisting of previously published genomes from various sources (Supplementary Data 9). Their taxonomic classification follows Baud et al. (2023), based on the same genome sequences.

The remaining 9 genomes were newly assembled or reassembled for this study. Specifically, isolates ESL0198, ESL0200, and ESL0170 had previously available Illumina-assembled genomes on NCBI (Supplementary Data 9); we improved these assemblies by incorporating additional Nanopore long-read data. The other six isolates - ESL0819, ESL0820, ESL0822, ESL0824, ESL0827 and ESL0825 - were cultured from *A. mellifera* gut homogenates on De Man, Rogosa and Sharpe (MRS) agar supplemented with 2% (w/v) fructose and 0.2% (w/v) L-

cysteine-HCl at 37 °C in anaerobic conditions. Their genome assemblies were generated de novo for this study.

All nine genomes were sequenced using a combination of Illumina MiSeq 100 bp paired-end (PE) reads and Oxford Nanopore MinION Mk1C long-reads. Illumina libraries were prepared using Nextera XT DNA library preparation at the Genomic Technologies Facility (GTF) of the University of Lausanne. Nanopore sequencing was conducted at the Institute for Infectious Diseases (Bern, Switzerland).

Nanopore reads were filtered for quality and length using Filtlong v0.2.1 (github repo: https://github.com/rrwick/Filtlong) with parameters --min_length 1000 --min_mean_q 10 --length_weight 10 --target_bases 400000000. Illumina reads were trimmed and quality-filtered using Trimmomatic v0.35[72] with settings PE ILLUMINA-CLIP:NexteraPE-PE.fa:2:30:10 LEADING:28 TRAILING:28 MINLEN:40.

Long-read assemblies were generated using Flye v2.9.1[76]. These assemblies were first polished by two rounds of mapping the Nanopore reads back to the assembly graph using GraphMap v0.5.2[77], followed of error correction with Racon v1.5.0[78]. Nanopre reads polishing was repeated for 2 iterations. Subsequently, three rounds of polishing with Illumina reads were performed using Bowtie2 v2.4.2[79] for mapping and Pilon v1.22[80] for error correction.

The final assemblies of these nine genomes (covering multiple *Bifidobacterium* strains) are available on the NCBI genome portal under the following BioSamples (Supplementary Data 9): SAMN49894004, SAMN49894003, SAMN49894002, SAMN49894001, SAMN4989 4000, SAMN49893999, SAMN49893998, SAMN49893997, SAMN4 9893996.

## Bacterial fraction assembly, binning and de-replication
Shotgun reads from the bacterial fraction were assembled using MetaSpades v3.15[81]. Then, contigs <1 kb were removed and the remaining contigs were binned independently for each sample into bacterial MAGs (bMAGs) using metabat2 v2.15[82]. The quality of the bMAGs was estimated using CheckM v1.0.13[83] and bMAGs with ≥75% completeness and <10% contamination were retained for subsequent analysis. The bMAGs and 220 bacterial genomes (Supplementary Data 3, 9) obtained from isolates from the gut of *A. mellifera* and other bee species[66] were clustered at 95% average nucleotide identity (ANI) into bacterial OTUs (bOTUs) using dRep v3.4.0[84]. The best genome for each bOTU according to dRep was designated as representative bMAGs. bMAGs were classified as the same species as the isolate genomes within the same bOTU, based on the taxonomical classification from Baud et al. (2023) (Supplementary Data 3, 9). If no isolate genome clustered with the bMAG, taxonomical classification was performed using GTDB-Tk v2.1.1[85] with the associated database release r207.

## Viral contigs identification and de-replication
Shotgun reads from the viral fraction were assembled using MetaSpades and MetaViralSpades v3.15[81]. The contigs resulting from both assemblers where combined, and viral contigs were predicted using VIBRANT v1.2.1[86], Virsorter2 v2.2.3[87], and viralVerify v1.1[88]. For each tool, contigs were assigned a confidence score from 1 to 3, where 3 denotes maximum confidence of the contig's viral origin. For contigs predicted by virsorter2, if its score were < 0.5, the contigs would get a score of 1; if its score were between 0.5 and 0.8, the contig would get a score of 2; otherwise, the contig would get a score of 3. For the contigs predicted by VIBRANT, if it were classified as "low quality draft", the contigs would get a score 1; if it was classified as "medium quality draft", the contigs would get a score of 2; otherwise, it would get a score of 3. For the contigs predicted by viralVerify, if its score were < 5, the contigs would get a score of 1; if its score were between 5 and 10, the contig would get a score of 2; otherwise, the contig would get a score of 3. When a contig was not identified by a tool, it would get a score of 0 for that tool. Finally, the contigs' scores were averaged, and the result was rounded to the closest integer to get a final score.

For each sample, contigs with a score ≥1 (identified at least with low confidence by all 3 tools) were retained. vMAGs of size < 10 kb or kmer frequency > 1.1 (contamination) were removed. The remaining vMAGs were quality checked using CheckV v1.0.1[89]. vMAGs classified as low-quality or higher were retained. In addition, vMAGs with contigs identified as 'lysogenic' by VIBRANT or as 'proviruses' by CheckV were classified as temperate, while the rest were classified as putative lytic.

FastANI v1.33[90] was used for all-vs-all vMAGs ANI and alignment fraction (AF) comparisons. Finally, vMAGs were clustered into viral OTUs (vOTUs) at 85% AF and 95% ANI[51] using dRep v3.4.0[84]. Two rounds of clustering were conducted: (1) average linkage clustering to create a database of phage contigs for read mapping, and (2) single linkage clustering to define the final set of vOTUs. For each vOTU, the longest vMAG with the highest CheckV quality score was chosen as the representative sequence. Only vOTUs containing at least one vMAG of quality medium or higher were retained for subsequent analyses.

## CRISPR-Cas analysis, spacer extraction, protospacer-to-spacer matching and genome homology inference
CRISPR-Cas loci were identified, and cas genes were subtyped from all bMAGs and isolate genomes using CRISPRCasFinder v4.2.20[91] and DefenseFinder v1.1.1[92], respectively. Only spacers from arrays with evidence level 4 according to CRISPRCasFinder were extracted for subsequent analysis. Moreover, to reduce arrays binned in the wrong bMAGs, spacers identified in bMAGs where < 10% of the bMAGs of the same bOTU contained a CRISPR-Cas locus were discarded. The recovered spacers were added to the CrisprOpenDB spacer database to increase the number of spacers available for protospacer-to-spacer matching. Alignments of spacers against medium- to complete-quality vMAGs were performed using blastn "blastn-short". Only matches with full query alignment and a maximum of 2 mismatches were considered as spacer-to-protospacer matches. Genome homology between vMAGs and bMAGs was inferred using FastANI v1.33[90]. Only matches with > 90% ANI and > 50% AF were considered. Moreover, matches with >80% alignment fraction and > 90% ANI were considered integrated prophages, and the vMAGs were classified as lysogenic (as in Johansen et al.[93]).

## Phage-bacteria interaction network analysis
A binary bipartite interaction network of medium- to high-quality vOTUs, as well as bMAGs and genomes from isolates of honeybee-associated genera was built in function of the historical interaction inferred through protospacer-to-spacer matching and genome homology inference. In addition, an analogous network was built in the same fashion but using only genomes from isolates.

Interaction modules (IMs) within the bipartite network were searched through the Label Propagation and Bipartite Recursively Induced Modules (LP-BRIM) algorithm[94] using the lpbrim v1 R package. This algorithm searches for the configuration of a bipartite network that maximizes its modularity (0 < Q ≤ 1, where a value of 1 denotes fully modular). Bacterial genomes were assigned to the modules to which most of the members of the same bOTU were assigned, and so were also the corresponding bOTUs. Genomes assigned to the bOTU *Gilliamella sp. cl-30_1* were initially placed in an IM by themselves by the lpbrim algorithm. However, given that this bOTU comprised highly similar genomes (median ANI = 99.9%) and shared vOTUs with other *Gilliamella* bOTUs, we manually reassigned this bOTU to the *Gilliamella_Frischella 1* IM. Similarly, genomes assigned to the bOTU *Bifidobacterium coryneforme* cl.−40_1 were initially placed in a separate IM. However, while one vOTU (vOTU_863) was assigned to this IM, all other vOTUs predicted to infect *Bifidobacterium coryneforme* cl.−40_1 were associated with the *Bifidobacterium_7* IM. Consequently, we manually reassigned this bOTU to the *Bifidobacterium_7* IM.

The nestedness of each module with more than one bacterial genome was measured using the "Nestedness metric based on Overlap

and Decreasing Fill" (NODF) metric[52]. All results were statistically tested against 1000 probabilistic degree null model networks that were generated as in Flores et al. (2013). These null model networks had the same number of rows and columns as the original module, as well as the same overall and marginal connections on average.

## Bacterial phylogeny

Gene calling for all bMAGs and isolate genomes was performed using Prodigal v2.6.3[95]. Then, OrthoFinder v2.2.7[96] -M msa option was used to compute a trimmed multiple sequence alignment (MSA) of 101 shared orthogroups protein sequences. Finally, a phylogeny of all bMAGs and isolated genomes was inferred from the trimmed MSA using IQtree v2.3.6 (Minh et al.[97]; -m LG + F + I + G4 -bb 1000).

## Viral protein sharedness network

Gene calling for all vMAGs was performed using Prodigal v2.6.3[95]. Medium- to high-quality vOTU were clustered according to their proteomes using vConTACT v2.0[98]. The resulting network was plotted using a custom R script.

To assess whether vOTU from the same IM tended to share similar genomic attributes, a previously described algorithm was used[26,99]. Briefly, for each node in the vConTACT network, a local neighborhood was defined. This neighborhood included all nodes that can be reached directly or indirectly through paths that fall within the first percentile of all pairwise path lengths between the focal node and all other nodes. Finally, the IM affiliations of all the nodes in the local neighborhood were compared to those of the focal node.

## Community profiling and SNV calling

Reads from the bacterial and viral fractions were mapped against a combined representative bOTUs database and a combined representative vOTUs database, respectively. The mapping was performed using Bowtie2 v2.5.1[79] in sensitive mode. Community profiling and read-mapping-based SNV calling were conducted using inStrain v2.5.1[100] in database mode. Only bOTUs with median sequencing coverage > 5x and breadth (proportion of bases covered by at least one read) > 0.5 were considered for subsequent analyses. Similarly, only vOTUs with breadth ≥ 0.7 were retained. Relative abundances were determined in function of the genome average coverage. For each sample, read-mapping-based nucleotide diversity for each position covered by at least 5 reads was calculated by inStrain[100] as follows:

$$\pi = 1 - \left( f(A)^2 + f(T)^2 + f(G)^2 + f(C)^2 \right) \quad (1)$$

where $f(A), f(T), f(G)$ and $f(C)$ denote the frequency of the four bases at the given position. Overall nucleotide diversity for each bOTUs was calculated by averaging the nucleotide diversity across the genome in each sample.

For each IM, Phylogenetic Diversity (PD) in a sample was calculated as the sum of the total phylogenetic branch length of all the bOTU present in that sample. This calculation was performed using the pd() function of the picante v1.8.2 R package. In each sample, bacterial within-module diversity for each IM was calculated as the average nucleotide diversity of all the bOTU in the given sample, multiplied by the PD. This measure was devised as a diversity measure that considers both species- and strain-level diversity.

## Community dissimilarity analysis

Jaccard dissimilarity matrices based on the presence/absence of genera, bOTUs, or vOTUs in each sample were computed using the vegan v2.6-4 R package. Only viral and bacterial genomes assigned to the core honeybee genera were considered for this computation. Moreover, inStrain[100] was used to calculate bacterial strain-level distances among samples. Briefly, the inStrain *compare* command was utilized to determine the population Average Nucleotide Identity (popANI; Olm et al., 2021) for each bOTU across samples. PopANI serves as a metric for ANI that considers both major and minor alleles within the same genome across the samples. Consequently, when comparing two samples, a higher popANI for a given bOTU indicates a greater similarity in the set of bacterial strains represented by that bOTU in these two samples. To quantify the differences in strain-level composition for a given bOTU between pairs of samples, a threshold of 99.9% popANI was used to determine the sharedness or not of the set of strains between two samples. Strain-level Jaccard distance between two samples was calculated as the fraction of bOTU not shared at the strain level between two samples. This resulted in a global distance that reflects strain-level dissimilarity across samples. Mantel tests between dissimilarity matrices were performed with the mantel() function of the vegan R package with 10,000 permutations. To ensure that the observed correlation between bacterial and viral dissimilarity matrices at the 99.9% popANI threshold reflected a broader trend, we repeated the analysis using a range of thresholds (96–99.99%) for defining strain-level similarity (Supplementary Fig. 7a).

To test the effect of the PBIN structure on community assembly, bacterial strain level and vOTU Jaccard dissimilarity matrix were computed by considering only phages and bacteria that belonged to a given IM. Then, the correlation between dissimilarity metrics were tested as through a Mantel test mentioned above. *P*-values were fdr adjusted for multiple testing. Only IMs where phages and bacteria were identified in at least 15 bees were considered for these analyses.

## Diversity correlations

Correlations between the various measurements of bOTU diversity and viral richness were tested with the cor.test() function of the *stats* R package or the stat_cor() function of the ggpubr v0.6.0 R package. Only IMs where phages and bacteria were identified in at least 15 bees were considered for these analyses.

To assess whether species PD or average nucleotide diversity explained viral richness variation across bees, the slope of the effect of PD and average nucleotide diversity on viral richness was inferred through a multiple robust linear model using the *rlm(viral_richness ~ PD + average_nucleotide_div)* function of the *MASS v7.3-60* R package. Significance of the slope with robust standard errors to account for heteroskedasticity of the data was tested using the coeftest() function of the *lmtest v0.9_4* R package.

## Reporting summary

Further information on research design is available in the Nature Portfolio Reporting Summary linked to this article.

## Data availability

Raw metagenomic data has been deposited to the NCBI Sequence Read Archive (SRA) under the Project ID PRJNA1232403. Newly sequenced bacterial genomes have been deposited to the NCBI genome portal under the Project ID PRJNA1232403 as the following BioSamples: SAMN49894004, SAMN49894003, SAMN49894002, SAMN49894001, SAMN49894000, SAMN49893999, SAMN49893998, SAMN49893997, SAMN49893996. The collection of bacterial metagenome-assembled genomes (bMAGs), isolate genomes, viral MAGs (vMAGs), as well as scripts, intermediate data files, and tables used to generate the figures in this manuscript are available in the Zenodo repository https://doi.org/10.5281/zenodo.16744037.

## Code availability

Code and details of parameters and software used are available at https://github.com/MalickNdiye/PHOSTER archived at https://doi.org/10.5281/zenodo.17087213[101].

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

## Acknowledgements

We thank the following people and organizations for supporting this project. Julien Marquis, Aline Charpagne, Melanie Dupasquier, and Johann Weber from the Lausanne Genomic Technologies Facility (GTF) for technical support during DNA extraction, library preparation, and sequencing. Vincent Sommerville for the feedback on the manuscript. This work was supported by an SNSF Project grant (grant ID 204369) and the NCCR Microbiomes, a National Centre of Competence in Research (grant numbers 180575 and 225148), funded by the Swiss National Science Foundation. All grants have been awarded to PE.

## Author contributions

Conceptualization – M.N., P.E., F.M. and G.B.R.; Methodology – M.N., P.E. and F.M.; Software – M.N.; Validation – M.N.; Formal analysis – M.N.; Investigation – M.N., P.E. and F.M.; Resources – P.E.; Data curation – M.N.; Writing – M.N., P.E. and F.M.; Original Draft – M.N., P.E. and F.M.; Review and Editing – M.N., F.M., P.E. and G.B.R.; Visualization – M.N.; Supervision – P.E.; Project administration – P.E.; Funding acquisition – P.E. and G.B.R.

## Competing interests

All authors declare no competing interests.
