## [Transparent Peer Review file · Nature Communications]

Phage diversity mirrors bacterial strain diversity in the honey bee gut microbiota

Corresponding Author: Professor Philipp Engel

Version 0:

Reviewer comments:

Reviewer #1

(Remarks to the Author)

The authors dissected the sequenced guts of 49 bees and intensively analyzed the bacteria-phage network. The idea is interesting. However, as the authors used gut tissue, it is impossible to distinguish the symbiont that the phages infect. Thus, this small sample size preliminary evidence is too weak to support the conclusion.

(Remarks on code availability)

Reviewer #2

(Remarks to the Author)

This manuscript provides a rigorous and well-executed analysis of phage-bacteria interactions in the honey bee gut, using paired metagenomics from 49 individual bees. The authors reconstruct a modular, nested PBIN and show that phage diversity most strongly reflects bacterial diversity at the strain level. The integration of CRISPR-spacer data, genome homology, and diversity metrics is methodologically sound and well-presented. However, I have a few major concerns as well as minor comments that should be addressed prior to this work being considered for publication.

Major Comments:

- 1) Clarity on resolution of strain-level profiling and SNV cutoff (Lines ~270–300, 840–860): The use of 99.9% popANI as a strain-level cutoff is appropriate, but further justification would strengthen the rationale. For example, were alternative thresholds tested (e.g., 99.5%, 99.95%) and shown to be less predictive of phage diversity correlations? Presenting a sensitivity analysis in a supplementary figure would enhance confidence in this threshold.
- 2) Potential biases in CRISPR-based host assignment (Lines ~245–265): While the authors acknowledge the limitations of CRISPR-based predictions, more discussion is warranted about the potential biases introduced by missing CRISPR-Cas systems in some genera (e.g., *Snodgrassella*, *Frischella*, *Bombilactobacillus*). Could this result in underrepresentation of phages linked to these bacteria? Do these gaps impact conclusions about module-specific correlations?
- 3) The definition used for “high-quality MAGs” are not in alignment with international MIMAG standard definition (Lines 106–107, Lines 410–423, and Table S3). The authors state that 478 high-quality MAGs were obtained and describe such MAGs as those with >75% completeness and <10% contamination. However, the MIMAG definition for high-quality is >90% completeness and <5% contamination (see Bowers et al. article: <https://www.nature.com/articles/nbt.3893>). Thus, the number of HQ MAGs should be recalculated with the proper cutoffs. Further, the abbreviation bMAGs (presumably relating to ‘bacterial MAGs’) should be defined here, and whether the high-quality MAGs are specifically being referred to.
- 4) Data availability issue (Line 108, Lines 588–591, Table S3): On line 108, the authors indicate 220 bacterial genomes were isolated from various bee species, which were used for comparison with MAGs from shotgun data. However, no accession numbers were given for any of these genomes in Table S3, and I was unable to locate any of the 220 isolate genomes anywhere in the Zenodo archive or the NCBI SRA BioProject ID PRJNA1232403. Each of these genome should be uploaded to NCBI genome portal and their accessions should be provided in Table S3. Moreover, a methods section should be added detailing how the genomes for the isolates were generated (e.g. Illumina short read? Or Nanopore long read?), the type of assembly software used (e.g., spades? Flye? Unicycler? Tricycler?), and other pertinent information. A separate

data table describing the status of the isolate genomes (draft, scaffold, complete/circularized) would also be helpful.

5) Nestedness interpretation and ecological significance (Lines ~260–280, 475–510): The nestedness pattern within modules is intriguing, but the mechanistic explanation is speculative. The authors might consider elaborating on whether this pattern supports a particular model of phage-host coevolution (e.g., arms race vs. fluctuating selection dynamics). Can any module-specific trends in host range breadth or generalist/specialist phage presence be shown?

6) Clarification of alpha diversity metric (Lines ~530–570): The "bacterial within-module diversity" metric, defined as phylogenetic diversity \times average nucleotide diversity, is innovative. However, readers unfamiliar with this compounded index may benefit from a more explicit conceptual justification. Was this multiplicative structure chosen based on theoretical considerations, or empirically optimized?

7) VLP vs. bacterial metagenome viral signal (Lines ~150–180, 720–740): The assertion that viral fractions enrich for lytic phages and bacterial fractions for prophages is important, but would benefit from clearer quantitative support. Could the authors report the proportions of lysogenic vs. lytic vMAGs per fraction? This could contextualize the strengths and limitations of using VLP-based approaches alone.

Minor Comments

Line 97: It should be clarified whether the two hives sampled from had identical background host genetics or not, as well as the rationale for sampling from only two colonies.

Line 195: The phrase "we obtained enough MAGs to capture most of the diversity..." is somewhat informal; consider rephrasing to "our dataset captured the majority of diversity observed in isolate genomes."

Line 223: The vOTU clustering thresholds (95% ANI, 85% AF) should be briefly justified, or referenced explicitly (e.g., Roux et al., 2019, if applicable).

Line 365–375: Clarify how vOTUs with no CRISPR or homology-based host predictions were treated in downstream network and diversity analyses. Were they excluded entirely?

Line 392: Please clarify whether genome homology was ever observed between phages and multiple bacterial genera (i.e., potential broad host range).

Line 423: The version of GTDB used (e.g. v95? v220?) should be specified here as well.

Line 675 (Figure 3 legend): "Dissimilarity distributions" could be replaced with "Jaccard dissimilarity distributions" for precision.

Line 883–885: In the nestedness section, the sentence beginning with "The two non-significant modules..." is somewhat repetitive and could be tightened for clarity.

(Remarks on code availability)

Reviewer #3

(Remarks to the Author)

Ndiaye and co-authors use metagenomic sequencing of the gut communities of individual honey bees to study the diversity of the bacteria and phages in this environment. Overall, they find the same sort of nested host-phage susceptibility pattern that has been determined by other methods in other systems, such as experimental tests of panels of phage isolates against different host strains. The other main finding is that more diverse phage communities are associated with more bacterial host diversity, within the phage-bacteria interaction modules, considering different host individuals. While both of these results are expected, it is striking and significant how they are able to reconstruct this pattern on multiple scales (between species groups and within them) from just metagenomic sequencing data. As they suggest, this could lead to further experimental studies that manipulate the bee gut communities to identify drivers.

I found the presentation, interpretation, and discussion of the data to be clear. They supported key conclusions with multiple different types of analyses.

The bioinformatic and statistical methods used were described in full detail. I did not note any flaws in the analysis. The authors share code, parameters, and data (both raw and processed) by depositing in NCBI and through a GitHub repository and Zenodo archive.

(Remarks on code availability)

Reviewer #4

(Remarks to the Author)

(Remarks on code availability)

Reviewer #5

(Remarks to the Author)

This is a very interesting study reporting that the interaction between phages and bacteria in the honeybee gut displays a highly modular structure as well as a high nestedness within each of the identified modules. More importantly, the viral and bacterial diversity values were high and correlated, especially at strain level, reflecting how different phages interact with different strains across species rather than with different species. These results highlight the importance monitoring phage-bacteria interactions at strain level to understanding the evolution of the microbial community structure and composition in natural ecosystems. Therefore:

- The results are novel and noteworthy, adding more evidence to the usefulness of using the microbial populations from the honeybee gut as a model to study host-microbe interactions.
- In the context of honeybee gut microbiome, the approach and work are original (as far as I know), thus the manuscript would be of very significance to the field and related fields.

Furthermore:

- The data support the conclusions and claims made and although there might be needed additional evidence to defend some of the conclusions, these are mere speculations that were presented as suggestions (last paragraph of the discussion)
- The methodology used is sound and very detailed, meeting the expected standards in the field, including an adequate analysis and the interpretation of the data.

Minor comments: when modules are explained/described, especially in the results section, it would be great if the authors indicate the specific bacterial group some of the modules refer to. For example, in line 178, what are the two non-significant modules? It can be observed in the figure, but text should also mention them. In this respect, I was wondering if the discussion can be expanded to explain more specifically the modularity divergency/similarity/consistency based on the bacterial genera. Only 3 genera -Frischella, Bartonella and Gilliamella- are discussed. How about the others (Snodgrassella, Commensalibacter, Bifidobacterium, Lactobacillus, Bombilactobacillus)?

(Remarks on code availability)

Reviewer #6

(Remarks to the Author)

The manuscript presents an interesting and timely study on phage-bacteria interactions in the honeybee gut. The data are substantial, and the analyses are appropriate. However, I have the following concerns and suggestions:

- I cannot find the phage sequences (vMAGs or vOTUs) shared with the manuscript. This severely limits my ability to review the paper, in particular to assess whether the host prediction approaches could be improved or validated. As it stands, the vast majority of host predictions are based on similarity, but phages are known for their extensive genetic diversity. Access to the sequences is essential.
- For a Nature Communications audience, I would expect a short introduction to the PBIN concept and why it matters. The authors describe well how the PBIN was built and analysed, but they assume that the reader is already familiar with microbial network concepts. A brief conceptual framing or figure would improve accessibility.
- The bioinformatic tools and methods used are what one would expect for this type of study.
- The code repository appears well maintained, though I did not clone or run it.

(Remarks on code availability)

Version 1:

Reviewer comments:

Reviewer #2

(Remarks to the Author)

Thank you for responding appropriately to my comments. I find your comments to be appropriate and contributing to a stronger manuscript altogether.

(Remarks on code availability)

Reviewer #4

(Remarks to the Author)

(Remarks on code availability)

Reviewer #6

(Remarks to the Author)

The authors have addressed all my comments.

(Remarks on code availability)

REVIEWER COMMENTS

Reviewer #1 (Remarks to the Author):

The authors dissected the sequenced guts of 49 bees and intensively analyzed the bacteria-phage network. The idea is interesting. However, as the authors used gut tissue, it is impossible to distinguish the symbiont that the phages infect. Thus, this small sample size preliminary evidence is too weak to support the conclusion.

Reply: We thank the reviewer for taking the time to evaluate our manuscript. We understand that the reviewer raises two major concerns: (1) the alleged impossibility of recovering phage–host interactions when sampling full guts, and (2) insufficient sampling size.

Regarding point (1), we are unclear as to why the presence of gut tissue in our samples would prevent the recovery of phage–host interaction for the following reasons:

1/ We would like to clarify that we did not sample only gut tissue but also the gut contents.

2/ Reads mapping to the honeybee genome (derived from the gut tissue) were removed prior to all downstream analyses (see *Methods*).

3/ We recovered several bacterial MAGs corresponding to species that are well-established members of the honeybee gut microbiota (Bonilla-Rosso & Engel, 2018; Kwong & Moran, 2016). Using genomic information from these bacterial MAGs (CRISPR spacers and genome homology) we were able to link them to the viral MAGs recovered from our samples. The fact that most viral MAGs were linked to core honeybee gut bacteria strongly suggests that our approach was effective and that these viruses are indeed interacting with bacteria from this ecosystem. Moreover, the naturally low complexity of the honey bee gut microbiota makes this approach more reliable than in more complex microbiomes such as soils and the human gut.

More fundamentally, it is unclear to us how the presence of gut tissue in the samples constitutes a methodological issue. This sampling approach is standard in the honeybee microbiota literature, including in studies focused on the virome (Bonilla-Rosso et al., 2020; Deboutte et al., 2020; Sbardellati & Vannette, 2024). Furthermore, the human gut virome is extensively studied using fecal samples and similar analytical methods (e.g., Camarillo-Guerrero et al., 2021; Lim et al., 2015; Lopez et al., 2025; Paez-

Espino et al., 2016; Shkoporov et al., 2019; Stern et al., 2012). From a bioinformatic perspective, we believe there is no fundamental difference between working with gut tissue and feces, as both inevitably contain some host DNA.

Regarding point (2), the reviewer appears to associate the need for increased sampling with the presence of gut tissue in our samples. We do not understand why working with gut samples would necessitate a larger sample size. The statistical analyses presented in the manuscript support the statistical significance of our findings, indicating that our current sampling provided sufficient statistical power to test our hypotheses. That said, we agree that future studies would benefit from a larger sample size, particularly from bees collected from more hives and diverse geographic locations. This would be essential both to corroborate our findings and to enable a more comprehensive characterization of the global honeybee gut virome. We have added a comment on this revised *Discussion* section:

Lines 365-369

Reviewer #2 (Remarks to the Author):

This manuscript provides a rigorous and well-executed analysis of phage-bacteria interactions in the honey bee gut, using paired metagenomics from 49 individual bees. The authors reconstruct a modular, nested PBIN and show that phage diversity most strongly reflects bacterial diversity at the strain level. The integration of CRISPR-spacer data, genome homology, and diversity metrics is methodologically sound and well-presented. However, I have a few major concerns as well as minor comments that should be addressed prior to this work being considered for publication.

Reply: We thank the reviewer for the concise and accurate summary of our work. We were happy to read that this reviewer found our study methodologically sound and well-presented.

In the following we have addressed the reviewer's comments point-by-point. However, please note that several line numbers referenced by the reviewer do not match those in our submitted document. We have done our best to interpret the intended sections and apologize for any potential misunderstandings.

Major Comments:

1) Clarity on resolution of strain-level profiling and SNV cutoff (Lines ~270–300, 840–860): The use of 99.9% popANI as a strain-level cutoff is appropriate, but further justification would strengthen the rationale. For example, were alternative thresholds tested (e.g., 99.5%, 99.95%) and shown to be less predictive of phage diversity correlations? Presenting a sensitivity analysis in a supplementary figure would enhance confidence in this threshold.

Reply: We thank the reviewer for this helpful suggestion. We agree that including a sensitivity analysis strengthens the rationale for our chosen 99.9% ANI threshold for strain level profiling. Therefore, we have now conducted an additional analysis testing the correlation between viral and bacterial beta-diversity across a range of within-species bacterial ANI thresholds, from 95% (species level) to 99.99%. This figure is reproduced below and included in the revised MS as Figure S7A.

This analysis showed a consistent increase in correlation strength (Mantel test, Spearman's rho) from 95% ANI (species-level) up to 99.93% ANI. The correlation strength plateaued between 99.7% and 99.93% ANI, then declined at higher thresholds, likely because too few strains are considered to be shared across samples at these very high ANI levels. While the highest correlation was observed at 99.8% ANI ($\rho = 0.49$), it was only marginally higher than at 99.9% ANI ($\rho = 0.46$), with no impact on the overall interpretation of our results.

Figure S7A: Correlation coefficients (Spearman's rho) between viral Jaccard compositional dissimilarity and bacterial Jaccard compositional dissimilarity across a range of ANI thresholds used to define strain-sharedness, from 95% (species level) to 99.99% ANI. The selected threshold of 99.9% ANI, used for strain-sharedness analyses in this study, is highlighted with a red box.

Given these findings, we have chosen to keep our original threshold of 99.9% ANI. In the revised manuscript, we have added the sensitivity analysis as Figure S7A (raw correlation data now available in Table S10) and now explicitly reference it in the *Results* and *Methods* sections to further support our threshold choice

Lines 234, 625-628

2) Potential biases in CRISPR-based host assignment (Lines ~245–265): While the authors acknowledge the limitations of CRISPR-based predictions, more discussion is warranted about the potential biases introduced by missing CRISPR-Cas systems in some genera (e.g., *Snodgrassella*, *Frischella*, *Bombilactobacillus*). Could this result in underrepresentation of phages linked to these bacteria? Do these gaps impact conclusions about module-specific correlations?

Reply: We thank the reviewer for this suggestion and agree with the point raised. The absence of CRISPR-Cas systems in certain bacterial genera may indeed lead to an underrepresentation of phages associated with these hosts. This limitation likely affects the host prediction outcomes and, consequently, the strength of observed within-module correlations in alpha and beta

diversity. In fact, we find that modules lacking CRISPR-Cas systems tend to exhibit weaker and no statistically significant diversity and composition correlations (see Fig. S7B and Fig. S9), which may be due to our reduced ability to recover the full diversity of their associated phages.

We agree that our previous MS did not acknowledge this point well enough. Therefore, we have now expanded the *Discussion* in the revised manuscript to clearly acknowledge this limitation.

Lines 354-358

3) The definition used for “high-quality MAGs” are not in alignment with international MIMAG standard definition (Lines 106-107, Lines 410-423, and Table S3). The authors state that 478 high-quality MAGs” were obtained and describe such MAGs as those with >75% completeness and <10% contamination. However, the MIMAG definition for high-quality is >90% completeness and <5% contamination (see Bowers et al. article: <https://www.nature.com/articles/nbt.3893>). Thus, the number of HQ MAGs should be recalculated with the proper cutoffs. Further, the abbreviation bMAGs (presumably relating to ‘bacterial MAGs’) should be defined here, and whether the high-quality MAGs are specifically being referred to.

Reply: We thank the reviewer for pointing this out. Indeed, according to the MIMAG guidelines our MAGs are a mix of high- and medium-quality MAGs. We chose to retain the same threshold ($\geq 75\%$ completeness and $< 10\%$ contamination) for MAG inclusion, as incorporating medium-quality MAGs does not compromise the conclusions of our study. As shown in Fig. S4, the number of spacers recovered from medium- to high-quality bMAGs does not significantly differ from that of isolate genomes from the same genus. Moreover, Figure S5 shows that a PBIN obtained only with isolate genomes has the same module-nested structure as the PBIN that includes medium- to high-quality bMAGs. This suggests that the inclusion of these MAGs did not introduce bias in phage–host linkage analysis. To incorporate this point in the revised MS:

1/ We now refer to these as "medium- to high-quality MAGs" to reflect their quality range (Line 119).

2/ We have also specified that 330 MAGs meet the MIMAG high-quality definition ($> 90\%$ completeness, $< 5\%$ contamination), while the rest falls into the medium-quality category (Line 105).

3/ We now define the abbreviation “bMAGs” (bacterial MAGs) clearly and clarify that this refers to all retained bMAGs, medium- or high-quality (Lines 104-107).

4) Data availability issue (Line 108, Lines 588-591, Table S3): On line 108, the authors indicate 220 bacterial genomes were isolated from various bee species, which were used for comparison with MAGs from shotgun data. However, no accession numbers were given for any of these genomes in Table S3, and I was unable to locate any of the 220 isolate genomes anywhere in the Zenodo archive or the NCBI SRA BioProject ID PRJNA1232403. Each of these genome should be uploaded to NCBI genome portal and their accessions should be provided in Table S3. Moreover, a methods section should be added detailing how the genomes for the isolates were generated (e.g. Illumina short read? Or Nanopore long read?), the type of assembly software used (e.g., spades? Flye? Unicycler? Tricycler?), and other pertinent information. A separate data table describing the status of the isolate genomes (draft, scaffold, complete/circularized) would also be helpful.

Reply: We thank the reviewer for pointing out this important oversight and apologize for the absence of these genomes in the original submission. Our bMAG dataset was supplemented with 220 bacterial isolate genomes, most of which were previously published and are available on NCBI (see Table S9 in the revised MS). However, a few isolates—specifically ESL0827, ESL0825, ESL0820, ESL0824, ESL0822, and ESL0819—were newly sequenced for this study using a combination of Illumina short-read and Nanopore long-read technologies. Additionally, while assemblies for ESL0198, ESL0200, and ESL0170 were already available on NCBI, we complemented them with Nanopore sequencing data to improve assembly quality.

All newly generated genomes have now been uploaded to the NCBI genome portal. In the revised manuscript, we have added Table S9, which provides detailed on how to recover each isolate genome, including its RefSeq, GeneBank and BioSample accession number and genome assembly status (e.g., contig, scaffold, complete), as reported by NCBI.

We have also added a new Methods section (*Bacterial Isolate Genomes Recovery, Sequencing and Assembly*, Lines 447-479) describing the origin of the isolates, the sequencing technologies used, and the genome assembly pipelines applied when assemblies were not already publicly available. Moreover, we have added all 220 bacteria isolate genomes to the Zotero repository linked to this study (<https://doi.org/10.5281/zenodo.16744037>). We hope this resolves the reviewer’s concerns and improves the transparency and reproducibility of our study.

5) Nestedness interpretation and ecological significance (Lines ~260–280, 475–510): The nestedness pattern within modules is intriguing, but the mechanistic explanation is speculative. The authors might consider elaborating on whether this pattern supports a particular model of phage-host coevolution (e.g., arms race vs. fluctuating selection dynamics). Can any module-specific trends in host range breadth or generalist/specialist phage presence be shown?

Reply: We thank the reviewer for this insightful comment. As we understand it, the reviewer is encouraging us to explore whether the nestedness patterns observed within modules of our PBIN could be explained by either evolutionary mechanisms or ecological factors, such as “*specific trends in host range breadth or generalist/specialist phage presence*”.

We do not think that our dataset is well-suited to investigate the evolutionary mechanisms giving rise to a nested interaction network. In the discussion (Line 285, unrevised MS), we had already noted that nestedness can arise from arms race dynamics between phages and bacteria. However, similar patterns can arise from fluctuating selection (Beckett & Williams, 2013; Borin et al., 2023; Hall et al., 2011; Holtappels et al., 2023; Koskella & Brockhurst, 2014). While fluctuating selection tends to promote long-term coexistence of phages and bacteria, arms race dynamics are expected to result in selective sweeps and high turnover of both phage and bacterial strains over time. Distinguishing between these processes would require longitudinal sampling of individual bees, which was beyond the scope of the present study. In response to this comment, we have revised the manuscript to replace the specific reference to “arms race” with the broader term “coevolution” to better reflect our agnosticism regarding the precise evolutionary processes driving the observed nestedness (Line 320).

From an ecological standpoint, studies on interaction networks in other systems (e.g., plant-pollinator networks) have suggested that nestedness can arise from differences in species prevalence (Krishna et al., 2008; Simmons et al., 2019; Vázquez et al., 2007). In such cases, generalist species tend to be more prevalent than specialists and thus have a higher likelihood of forming interactions. Applied to our system, this would imply that nestedness reflects broader distribution patterns of phages and bacterial strains. To test this, in the revised MS we correlated the host range of each vOTU (defined as the number of bacterial genomes it was predicted to interact with in Fig. 2B) with its prevalence across individual bees. We found a positive correlation between vOTU host range and prevalence (Fig. S6C, revised manuscript), supporting the hypothesis that prevalence contributes to nestedness in our PBIN.

Figure S6C: Correlation between vOTU prevalence across individual bees and host range, shown separately for each interaction module (IM). Host range is defined as the number of bacterial genomes with which a given vOTU was predicted to interact (Fig. 2B). Spearman's rho and p-values are indicated on each plot, along with a blue linear regression line for visual reference. Only IMs containing both phages and bacteria detected in at least 15 bees were included in this analysis.

Most of these correlations are primarily driven by a few highly prevalent and generalist vOTUs, and therefore should be interpreted with caution. Nonetheless, the consistency of this pattern across multiple IMs suggests it may reflect a genuine underlying mechanism. Importantly, this observation does not rule out the role of other processes, such as coevolutionary dynamics, in contributing to the observed nestedness.

In the revised manuscript, we now highlight this finding in the Results section (Lines 225-229) and expand upon its implications in the Discussion (Lines 322-329).

6) Clarification of alpha diversity metric (Lines ~530–570): The "bacterial within-module diversity" metric, defined as phylogenetic diversity \times average nucleotide diversity, is innovative. However, readers unfamiliar with this compounded index may benefit from a more explicit conceptual justification. Was this multiplicative structure chosen based on theoretical considerations, or empirically optimized?

Reply: We thank the reviewer for this insightful comment. This metric was devised to capture both species-level and strain-level diversity within each bacterial interaction module by combining two components: phylogenetic diversity (PD), which reflects the

evolutionary diversity of bacterial species within a module, and average nucleotide diversity, which quantifies genetic variation within species.

We chose a multiplicative structure to reflect the idea that each species—represented in the PD—contributes, on average, a certain amount of within-species (i.e., strain-level) diversity. Specifically, genome-wide nucleotide diversity was calculated for each species individually, then averaged across all species within a given interaction module. This average represents the mean contribution of each species to strain-level diversity. By multiplying it with PD, we obtain a composite metric that approximates the total genetic diversity within the module, integrating both taxonomic breadth and genetic depth.

We agree that the previous version of the MS was unclear on this point so we have now clarified this concept in the revised manuscript (Lines 245-253)

7) VLP vs. bacterial metagenome viral signal (Lines ~150–180, 720–740): The assertion that viral fractions enrich for lytic phages and bacterial fractions for prophages is important, but would benefit from clearer quantitative support. Could the authors report the proportions of lysogenic vs. lytic vMAGs per fraction? This could contextualize the strengths and limitations of using VLP-based approaches alone.

Reply: We thank the reviewer for this valuable suggestion. In response, we have now included the number of vMAGs classified as lytic (virulent) and lysogenic (temperate) in the Results section to provide clearer quantitative support for our conclusions:

“Lines 126-130: The bacterial and viral fractions contained a similar number of vMAGs classified as temperate phages, with 1’685 and 1’682 vMAGs, respectively (Table S4). In contrast, the viral fraction harbored more vMAGs classified as virulent phages (4’103) compared to the bacterial fraction (2’551; Table S4).”

Together with our previous observation that VLPs are enriched in the viral fraction, we believe this addition helps to clarify the advantages of VLP-based approaches, which are generally enriched for phage sequences (particularly lytic phages).

Minor Comments

Line 97: It should be clarified whether the two hives sampled from had identical background host genetics or not, as well as the rationale for sampling from only two colonies.

Reply: We thank the reviewer for this observation. We do not track the genetic background of our colonies, but it is likely that the two sampled hives had different genetic backgrounds. Moreover, we would like to point out that individual worker bees within a colony are **not** genetically identical, since queen bees are polyandrous and produce daughters of mixed paternity. However, evaluating the effect of the host genetics on the virome is beyond the scope of this paper. We chose to sample from more than one colony to reduce the likelihood that our findings were driven by colony-specific effects. The paired sampling of viromes and prokaryomes from individual honeybees is extremely laborious and, as a first study employing this technique, we deem our sampling satisfactory. Nonetheless, in the revised discussion, we now emphasize the need for future research to include a broader set of colonies spanning diverse geographic locations. This would be essential both to corroborate our findings and to enable a more comprehensive characterization of the global honeybee gut virome (see Lines 365-369).

Line 195: The phrase “we obtained enough MAGs to capture most of the diversity...” is somewhat informal; consider rephrasing to “our dataset captured the majority of diversity observed in isolate genomes.”

Reply: Thanks for your suggestion, we rephrased this sentence accordingly (Lines 121-122).

Line 223: The vOTU clustering thresholds (95% ANI, 85% AF) should be briefly justified, or referenced explicitly (e.g., Roux et al., 2019, if applicable).

Reply: Thanks for your suggestion. Our clustering was indeed based on the paper you cite. We now added the citation in the revised manuscript, both in the *Results* and *Methods* section (Lines 135, 518, 836).

Line 365–375: Clarify how vOTUs with no CRISPR or homology-based host predictions were treated in downstream network and diversity analyses. Were they excluded entirely?

Reply: We thank the reviewer for pointing this out. vOTUs without CRISPR- or homology-based host predictions were excluded from downstream network and diversity analyses, as our analytical framework relied on confidently assigning a bacterial host to each phage. Similarly, we also excluded vOTUs assigned to non-core members of the honeybee gut microbiome, as our focus was specifically on the core microbial community. This has now been clarified in the revised manuscript (Lines 154-157).

Line 392: Please clarify whether genome homology was ever observed between phages and multiple bacterial genera (i.e., potential broad host range).

Reply: We thank the reviewer for raising this point. We investigated this, and out of the 1,069 vOTUs recovered in this study, only 23 exhibited genome homology with bacteria from more than one genus. Of these, 16 vOTUs showed genome homology with bacteria from the genera *Frischella* and *Gilliamella*, which is consistent with their clustering within the same IM. The remaining 7 vOTUs displayed genome homology with *Frischella*, *Gilliamella*, and *Snodgrassella*. This is also biologically plausible, given the relatively close phylogenetic relationship among these genera (all Proteobacteria) and the fact that they co-localize in the ileum forming multispecies biofilms.

Overall, the number of phages exhibiting cross-genus genome homology was very low and can be considered negligible in the context of our analyses. We note that the line referenced in the reviewer's comment does not appear to align with the submitted version of the manuscript. Thus, we have included this information in the revised version at Lines 176-183.

Line 423: The version of GTDB used (e.g. v95? v220?) should be specified here as well.

Reply: Unfortunately, the line referenced in this comment does not appear to correspond to the correct location in the manuscript, and we were unable to determine the exact passage being referred to. However, we would like to clarify that the version of GTDB-Tk used throughout the study is specified in the *Methods* section (Line 492; GTDB-Tk v2.1.1). Moreover, in the revised manuscript, we added the database release number (r207) in the same result section (Line 493).

Line 675 (Figure 3 legend): "Dissimilarity distributions" could be replaced with "Jaccard dissimilarity distributions" for precision.

Reply: We thank the reviewer for this suggestion. However, the legend for Figure 3 (panels C and D) already specifies "*Jaccard dissimilarity distribution*." In the revised legend of Figure 3, when we refer to "*compositional dissimilarity*", we now specify "*Jaccard compositional dissimilarity*" as we suspect the reviewer may have been referring to this wording, given the similarity of the terms.

Line 883–885: In the nestedness section, the sentence beginning with “The two non-significant modules...” is somewhat repetitive and could be tightened for clarity.

Reply: We thank the reviewer for this suggestion. We agree that the sentence could be tightened for clarity. However, in response to a related comment from Reviewer #5, we decided to explicitly name the two non-significant modules to improve clarity. The revised passage now reads:

“Lines 191-196: [...] revealing that seven (Gilliamella_Frischella_1, Snodgrassella_3, Bartonella_5, Bartonella_6, Bifidobacterium_7, Lactobacillus_8, and Bombilactobacillus_9) of the nine IMs were significantly nested (Fig. 2D). The Snodgrassella_3 and Commensalibacter_4 IMs had the fewest phage–bacteria pairs, which may explain why they did not exhibit statistically significant nestedness. “

We hope this revision satisfactorily addresses the concerns of both Reviewer #2 and Reviewer #5.

Reviewer #3 (Remarks to the Author):

Ndiaye and co-authors use metagenomic sequencing of the gut communities of individual honey bees to study the diversity of the bacteria and phages in this environment. Overall, they find the same sort of nested host-phage susceptibility pattern that has been determined by other methods in other systems, such as experimental tests of panels of phage isolates against different host strains. The other main finding is that more diverse phage communities are associated with more bacterial host diversity, within the phage-bacteria interaction modules, considering different host individuals. While both of these results are expected, it is striking and significant how they are able to reconstruct this pattern on multiple scales (between species groups and within them) from just metagenomic sequencing data. As they suggest, this could lead to further experimental studies that manipulate the bee gut communities to identify drivers.

I found the presentation, interpretation, and discussion of the data to be clear. They supported key conclusions with multiple different types of analyses.

The bioinformatic and statistical methods used were described in full detail. I did not note any flaws in the analysis. The

authors share code, parameters, and data (both raw and processed) by depositing in NCBI and through a GitHub repository and Zenodo archive.

Reply: We thank the reviewer for their very positive feedback!

Reviewer #4 (Remarks to the Author):

Reply: Okay!

Reviewer #5 (Remarks to the Author):

This is a very interesting study reporting that the interaction between phages and bacteria in the honeybee gut displays a highly modular structure as well as a high nestedness within each of the identified modules. More importantly, the viral and bacterial diversity values were high and correlated, especially at strain level, reflecting how different phages interact with different strains across species rather than with different species. These results highlight the importance monitoring phage-bacteria interactions at strain level to understanding the evolution of the microbial community structure and composition in natural ecosystems. Therefore:

- The results are novel and noteworthy, adding more evidence to the usefulness of using the microbial populations from the honeybee gut as a model to study host-microbe interactions.
- In the context of honeybee gut microbiome, the approach and work are original (as far as I know), thus the manuscript would be of very significance to the field and related fields.

Furthermore:

- The data support the conclusions and claims made and although there might be needed additional evidence to defend some of the conclusions, these are mere speculations that were presented as suggestions (last paragraph of the discussion)
- The methodology used is sound and very detailed, meeting the expected standards in the field, including an adequate analysis and the interpretation of the data.

Minor comments: when modules are explained/described, especially in the results section, it would be great if the authors indicate the specific bacterial group some of the modules refer to. For example, in line 178, what are the two non-significant modules? It can be observed in the figure, but text should also mention them. In this respect, I was wondering if the discussion can be expanded to explain more specifically the modularity divergency/similarity/consistency based on the bacterial genera. Only 3 genera -*Frischella*, *Bartonella* and *Gilliamella*- are discussed. How about the others (*Snodgrassella*, *Commensalibacter*, *Bifidobacterium*, *Lactobacillus*, *Bombilactobacillus*)?

Reply: We thank the reviewer for this helpful suggestion. In the revised manuscript, we have now added the names of the specific modules directly in the text when referring to them (Lines 191-196, 262-266), to make it easier for readers to follow without needing to refer to the figures.

In addition, we have expanded the discussion to include a more comprehensive interpretation of modularity in relation to bacterial phylogeny. Beyond *Frischella*, *Bartonella*, and *Gilliamella*, we now also discuss the patterns observed in modules associated with *Snodgrassella*, *Commensalibacter*, *Bifidobacterium*, *Lactobacillus*, and *Bombilactobacillus*. This addition offers a more complete interpretation of the ecological and evolutionary patterns underlying the observed network structure. The relevant text can be found on Lines 290-314.

Reviewer #6 (Remarks to the Author):

The manuscript presents an interesting and timely study on phage-bacteria interactions in the honeybee gut. The data are substantial, and the analyses are appropriate. However, I have the following concerns and suggestions:

- I cannot find the phage sequences (vMAGs or vOTUs) shared with the manuscript. This severely limits my ability to review the paper, in particular to assess whether the host prediction approaches could be improved or validated. As it stands, the vast

majority of host predictions are based on similarity, but phages are known for their extensive genetic diversity. Access to the sequences is essential.

We thank the reviewer for highlighting this important point and apologize for the oversight. The fasta files of the 10,021 vMAGs recovered in our analysis are now publicly available in the *vMAGs.zip* folder within the updated version of the Zenodo repository associated with this study (<https://doi.org/10.5281/zenodo.16744037>).

Additionally, we thank the reviewer for raising this important point regarding the phage–host linkage methods used in our study. As correctly noted, our approach relied on CRISPR spacer–protospacer matching and genome homology between phages and bacterial genomes. While these methods are indeed based on nucleotide similarity, they remain among the most precise and widely accepted approaches for linking viruses to specific bacterial hosts (Dion et al., 2021; Johansen et al., 2022; Stern et al., 2012). We were able to assign approximately 75% of phages to a bacterial host, enabling the construction of a phage–bacteria interaction network (PBIN) with clear phylogenetic structure (Fig. 2A, 2C) and ecological relevance (Fig. 3–4).

We acknowledge that alternative tools such as *iPHoP* (Roux et al., 2023) or *VirHostMatcher* (Wang et al., 2020) incorporate additional features like oligonucleotide usage, codon bias, or machine learning–based host predictions. However, these tools are typically optimized to predict host identity at broader taxonomic ranks (e.g., genus or family), rather than at the resolution of individual bacterial genomes, and are less suited for the construction of interaction networks requiring high-resolution linkages. In our case, the relatively low complexity and well-characterized composition of the honey bee gut microbiota provided a unique context in which sequence-similarity based methods could achieve an unusually high host recovery rate, outperforming their typical performance in complex environments such as soil or the human gut. Thus, while we recognize the value of complementary host prediction tools, we opted to prioritize high-resolution, direct sequence-based approaches to ensure accurate host assignments to evaluate the topology of the PBIN.

- For a Nature Communications audience, I would expect a short introduction to the PBIN concept and why it matters. The authors describe well how the PBIN was built and analysed, but they assume that the reader is already familiar with microbial network concepts. A brief conceptual framing or figure would improve accessibility.

We thank the reviewer for this helpful suggestion and agree that introducing the concept of phage–bacteria interaction networks (PBINs) and their relevance would improve accessibility for a broader audience. In response, we have revised the relevant section in the manuscript’s introduction to provide a concise conceptual framing of PBINs and to explain their ecological importance. The revised text now reads:

“Lines 46–60: Second, many ecological surveys have not considered the structure of the phage-bacteria interaction networks (PBIN), which map infection relationships between phages and their bacterial hosts (i.e. which phage infect which host). Interaction networks are powerful tools for analyzing complex ecological systems and have been widely used in other domains of ecology, such as food webs, plant–pollinator systems, and microbial communities (Bascompte et al., 2003; Karimi et al., 2017; Pascual & Dunne, 2005). Analyzing the topology of these networks can reveal non-random structures such as modules, i.e. clusters of interacting species that are more connected to each other than to the rest of the network. These modules often represent groups with shared ecological or evolutionary characteristics, thereby simplifying community complexity and enabling a better understanding of local dynamics and ecological patterns (Gibson et al., 2023; Olesen et al., 2007). Both theoretical and experimental studies show that PBIN are often modular, i.e. interactions occur within specific groups of phages and bacteria, with little overlap among groups (Flores et al., 2011; Holtappels et al., 2023; Kauffman et al., 2022; Pratama & van Elsas, 2018; Weitz et al., 2013). Therefore, associations between phage and bacterial diversity are expected to be stronger within modules rather than between modules, but this has not been explored so far. “

- The bioinformatic tools and methods used are what one would expect for this type of study.
- The code repository appears well maintained, though I did not clone or run it.

Reply: We are happy to read that the reviewer found our study methodologically sound and the code repository well maintained.

REFERENCES:

Bascompte, J., Jordano, P., Melián, C. J., & Olesen, J. M. (2003). The nested assembly of plant–animal mutualistic networks.

Proceedings of the National Academy of Sciences, 100(16), 9383–9387. <https://doi.org/10.1073/pnas.1633576100>

Beckett, S. J., & Williams, H. T. P. (2013). Coevolutionary diversification creates nested-modular structure in phage-bacteria

interaction networks. *Interface Focus*, 3(6), 20130033. <https://doi.org/10.1098/rsfs.2013.0033>

- Bonilla-Rosso, G., & Engel, P. (2018). Functional roles and metabolic niches in the honey bee gut microbiota. *Current Opinion in Microbiology*, 43, 69–76. <https://doi.org/10.1016/j.mib.2017.12.009>
- Bonilla-Rosso, G., Steiner, T., Wichmann, F., Bexkens, E., & Engel, P. (2020). Honey bees harbor a diverse gut virome engaging in nested strain-level interactions with the microbiota. *Proceedings of the National Academy of Sciences*, 117(13), 7355–7362. <https://doi.org/10.1073/pnas.2000228117>
- Borin, J. M., Lee, J. J., Lucia-Sanz, A., Gerbino, K. R., Weitz, J. S., & Meyer, J. R. (2023). *Rapid bacteria-phage coevolution drives the emergence of multi-scale networks* (p. 2023.04.13.536812). bioRxiv. <https://doi.org/10.1101/2023.04.13.536812>
- Camarillo-Guerrero, L. F., Almeida, A., Rangel-Pineros, G., Finn, R. D., & Lawley, T. D. (2021). Massive expansion of human gut bacteriophage diversity. *Cell*, 184(4), Article 4. <https://doi.org/10.1016/j.cell.2021.01.029>
- Deboutte, W., Beller, L., Yinda, C. K., Maes, P., de Graaf, D. C., & Matthijnssens, J. (2020). Honey-bee-associated prokaryotic viral communities reveal wide viral diversity and a profound metabolic coding potential. *Proceedings of the National Academy of Sciences*, 117(19), 10511–10519. <https://doi.org/10.1073/pnas.1921859117>
- Dion, M. B., Plante, P.-L., Zufferey, E., Shah, S. A., Corbeil, J., & Moineau, S. (2021). Streamlining CRISPR spacer-based bacterial host predictions to decipher the viral dark matter. *Nucleic Acids Research*, 49(6), Article 6. <https://doi.org/10.1093/nar/gkab133>

Flores, C. O., Meyer, J. R., Valverde, S., Farr, L., & Weitz, J. S. (2011). Statistical structure of host–phage interactions.

Proceedings of the National Academy of Sciences, 108(28), E288–E297. <https://doi.org/10.1073/pnas.1101595108>

Gibson, T. E., Kim, Y., Acharya, S., Kaplan, D. E., DiBenedetto, N., Lavin, R., Berger, B., Allegretti, J. R., Bry, L., & Gerber, G. K.

(2023). *Microbial dynamics inference at ecosystem-scale* (p. 2021.12.14.469105). bioRxiv.

<https://doi.org/10.1101/2021.12.14.469105>

Hall, A. R., Scanlan, P. D., Morgan, A. D., & Buckling, A. (2011). Host–parasite coevolutionary arms races give way to fluctuating

selection. *Ecology Letters*, 14(7), 635–642. <https://doi.org/10.1111/j.1461-0248.2011.01624.x>

Holtappels, D., Alfenas-Zerbini, P., & Koskella, B. (2023). Drivers and consequences of bacteriophage host range. *FEMS*

Microbiology Reviews, 47(4), fuad038. <https://doi.org/10.1093/femsre/fuad038>

Johansen, J., Plichta, D. R., Nissen, J. N., Jespersen, M. L., Shah, S. A., Deng, L., Stokholm, J., Bisgaard, H., Nielsen, D. S.,

Sørensen, S. J., & Rasmussen, S. (2022). Genome binning of viral entities from bulk metagenomics data. *Nature*

Communications, 13(1), Article 1. <https://doi.org/10.1038/s41467-022-28581-5>

Karimi, B., Maron, P. A., Chemidlin-Prevost Boure, N., Bernard, N., Gilbert, D., & Ranjard, L. (2017). Microbial diversity and

ecological networks as indicators of environmental quality. *Environmental Chemistry Letters*, 15(2), 265–281.

<https://doi.org/10.1007/s10311-017-0614-6>

Kauffman, K. M., Chang, W. K., Brown, J. M., Hussain, F. A., Yang, J., Polz, M. F., & Kelly, L. (2022). Resolving the structure of phage–bacteria interactions in the context of natural diversity. *Nature Communications*, *13*(1), Article 1.

<https://doi.org/10.1038/s41467-021-27583-z>

Koskella, B., & Brockhurst, M. A. (2014). Bacteria-phage coevolution as a driver of ecological and evolutionary processes in microbial communities. *FEMS Microbiology Reviews*, *38*(5), 916–931. <https://doi.org/10.1111/1574-6976.12072>

Krishna, A., Guimarães Jr, P. R., Jordano, P., & Bascompte, J. (2008). A neutral-niche theory of nestedness in mutualistic networks. *Oikos*, *117*(11), 1609–1618. <https://doi.org/10.1111/j.1600-0706.2008.16540.x>

Kwong, W. K., & Moran, N. A. (2016). Gut microbial communities of social bees. *Nature Reviews Microbiology*, *14*(6), Article 6.

<https://doi.org/10.1038/nrmicro.2016.43>

Lim, E. S., Zhou, Y., Zhao, G., Bauer, I. K., Droit, L., Ndao, I. M., Warner, B. B., Tarr, P. I., Wang, D., & Holtz, L. R. (2015). Early life dynamics of the human gut virome and bacterial microbiome in infants. *Nature Medicine*, *21*(10), 1228–1234.

<https://doi.org/10.1038/nm.3950>

Lopez, J. A., McKeithen-Mead, S., Shi, H., Nguyen, T. H., Huang, K. C., & Good, B. H. (2025). Abundance measurements reveal the balance between lysis and lysogeny in the human gut microbiome. *Current Biology*, *35*(10), 2282-2294.e11.

<https://doi.org/10.1016/j.cub.2025.03.073>

- Olesen, J. M., Bascompte, J., Dupont, Y. L., & Jordano, P. (2007). The modularity of pollination networks. *Proceedings of the National Academy of Sciences*, 104(50), 19891–19896. <https://doi.org/10.1073/pnas.0706375104>
- Paez-Espino, D., Eloë-Fadrosch, E. A., Pavlopoulos, G. A., Thomas, A. D., Huntemann, M., Mikhailova, N., Rubin, E., Ivanova, N. N., & Kyrpides, N. C. (2016). Uncovering Earth’s virome. *Nature*, 536(7617), 425–430. <https://doi.org/10.1038/nature19094>
- Pascual, M., & Dunne, J. A. (2005). *Ecological Networks: Linking Structure to Dynamics in Food Webs*. Oxford University Press.
- Pratama, A. A., & van Elsas, J. D. (2018). The ‘Neglected’ Soil Virome – Potential Role and Impact. *Trends in Microbiology*, 26(8), 649–662. <https://doi.org/10.1016/j.tim.2017.12.004>
- Roux, S., Camargo, A. P., Coutinho, F. H., Dabdoub, S. M., Dutilh, B. E., Nayfach, S., & Tritt, A. (2023). iPHoP: An integrated machine learning framework to maximize host prediction for metagenome-derived viruses of archaea and bacteria. *PLOS Biology*, 21(4), e3002083. <https://doi.org/10.1371/journal.pbio.3002083>
- Sbardellati, D. L., & Vannette, R. L. (2024). Targeted viromes and total metagenomes capture distinct components of bee gut phage communities. *Microbiome*, 12(1), 155. <https://doi.org/10.1186/s40168-024-01875-0>
- Shkoporov, A. N., Clooney, A. G., Sutton, T. D. S., Ryan, F. J., Daly, K. M., Nolan, J. A., McDonnell, S. A., Khokhlova, E. V., Draper, L. A., Forde, A., Guerin, E., Velayudhan, V., Ross, R. P., & Hill, C. (2019). The Human Gut Virome Is Highly Diverse, Stable, and Individual Specific. *Cell Host & Microbe*, 26(4), 527-541.e5. <https://doi.org/10.1016/j.chom.2019.09.009>

- Simmons, B. I., Vizentin-Bugoni, J., Maruyama, P. K., Cotton, P. A., Marín-Gómez, O. H., Lara, C., Rosero-Lasprilla, L., Maglianesi, M. A., Ortiz-Pulido, R., Rocca, M. A., Rodrigues, L. C., Tinoco, B. A., Vasconcelos, M. F., Sazima, M., Martín González, A. M., Sonne, J., Rahbek, C., Dicks, L. V., Dalsgaard, B., & Sutherland, W. J. (2019). Abundance drives broad patterns of generalisation in plant–hummingbird pollination networks. *Oikos*, *128*(9), 1287–1295. <https://doi.org/10.1111/oik.06104>
- Stern, A., Mick, E., Tirosh, I., Sagy, O., & Sorek, R. (2012). CRISPR targeting reveals a reservoir of common phages associated with the human gut microbiome. *Genome Research*, *22*(10), Article 10. <https://doi.org/10.1101/gr.138297.112>
- Vázquez, D. P., Melián, C. J., Williams, N. M., Blüthgen, N., Krasnov, B. R., & Poulin, R. (2007). Species abundance and asymmetric interaction strength in ecological networks. *Oikos*, *116*(7), 1120–1127. <https://doi.org/10.1111/j.0030-1299.2007.15828.x>
- Wang, W., Ren, J., Tang, K., Dart, E., Ignacio-Espinoza, J. C., Fuhrman, J. A., Braun, J., Sun, F., & Ahlgren, N. A. (2020). A network-based integrated framework for predicting virus–prokaryote interactions. *NAR Genomics and Bioinformatics*, *2*(2), lqaa044. <https://doi.org/10.1093/nargab/lqaa044>
- Weitz, J. S., Poisot, T., Meyer, J. R., Flores, C. O., Valverde, S., Sullivan, M. B., & Hochberg, M. E. (2013). Phage–bacteria infection networks. *Trends in Microbiology*, *21*(2), 82–91. <https://doi.org/10.1016/j.tim.2012.11.003>